# Influence of intensive agriculture and geological heterogeneity on the recharge of an arid aquifer system (Saq-Ram, Arabian Peninsula) inferred from GRACE data.

Pierre Seraphin [a], Julio Gonçalvès [a], Bruno Hamelin [a], Thomas Stieglitz [a], Pierre Deschamps [a]

[a] Aix Marseille Université, CNRS, IRD, Coll France, INRAE, CEREGE UM 34, 13545 Aix en Provence, France

*Correspondence to*: Pierre Seraphin (seraphin@cerege.fr)

## Keywords

Saq-Ram Aquifer System, Satellite-gravity data, Water Budget, Spatial distribution of the recharge, Irrigation return flow

## Abstract

This study assesses the detailed water budget of the Saq-Ram Aquifer System (520 000 km²) over the 2002-2019 period using satellite-gravity data from the Gravity Recovery And Climate Experiment (GRACE). The three existing GRACE solutions were tested for their local compatibility to compute groundwater storage variations in combination with the three soil moisture datasets available from the Global Land Data Assimilation System (GLDAS) land surface models. Accounting for groundwater pumping, artificial recharge and natural discharge uniformly distributed over the Saq-Ram domain, the GRACE-derived water mass balance calculation yields a long-term estimate of the domain-averaged natural recharge of $(2.4 \pm 1.4)$ mm yr$^{-1}$, corresponding to $(4.4 \pm 2.6)$ % of the annual average rainfall.

Beyond the regional-scale approach proposed here, spatial heterogeneities regarding the groundwater recharge were identified. The first source of heterogeneity is of anthropogenic origin. Within agricultural plots, irrigation excess is great enough to artificially recharge the aquifer (i.e. $(167 \pm 83)$ mm yr$^{-1}$ distributed over irrigated areas). However, on the outskirts of these crop areas subjected only to the natural recharge but still influenced by pumping drawdown, there is a risk of relative disconnection from the infiltration front with the declining water table (i.e. the unsaturated zone thickens faster than percolation flows through it), making effective recharge locally zero. The second source of recharge heterogeneity identified here is natural: Volcanic lava deposits (called Harrats on the Arabian Peninsula) which cover 8 % of the Saq-Ram Aquifer domain but contribute to more than 50 % of the total natural recharge. Hence, in addition to this application on the Arabian Peninsula, this study strongly indicates a major control of geological context on arid aquifer recharge which has been poorly discussed hitherto.

## 1. Introduction

Freshwater resources in arid regions of the world face growing pressure. Limited reserves, sporadic rainfall, droughts, agricultural production, increasing population and living standards are contributing to environmental and economic pressures. As defined by Gleeson et al. (2020): "groundwater sustainability is maintaining long-term, dynamically stable storage and flows of high-quality groundwater using inclusive, equitable, and long-term governance and
management". Groundwater resources in arid zones have been heavily exploited for the past 50 years or so, in order to meet growing demands, which has led to overexploitation and local long-term depletion in many cases (Al-Zyoud et al., 2015; Othman et al., 2018). When aquifer recharge is much lower than withdrawals, this depletion can constitute permanent groundwater mining (Bierkens and Wada, 2019; Wada et al., 2010). In arid and semi-arid regions, this is a
frequent phenomenon, in particular where large aquifer replenishment mostly occurred under past climatic conditions (so called "fossil aquifers").

    Shared between Jordan and Saudi Arabia, the Saq-Ram Aquifer System ($0.5 \times 10^6$ km²) is the main water resource of the region, the exploitation of which has enabled the development of intensive irrigated agriculture since the mid-1980s. This multi-layered aquifer system is part
of the larger Arabian Aquifer System which has been recognized as one of the two most overstressed systems in the world, presenting the highest depletion rates combined with the lowest available recharges (Richey et al., 2015). Despite having significant groundwater reserves, Saudi Arabia had to cope with these difficulties by abandoning its goal of cereal self-sufficiency when groundwater mining became evident (Konikow and Kendy, 2005).

Groundwater recharge is a pivotal term of an aquifer's water mass-balance when it comes to assessing the (un)sustainability of its exploitation. Recharge can be assessed indirectly or directly (Banks et al., 2021; Shanafield and Cook, 2014) by studying either surface water bodies (e.g. seepage meters, baseflow discharge, heat tracers), unsaturated (e.g. lysimeters) or saturated zones (e.g. Water-table Fluctuation). Various types of estimation methods
(MacDonald et al., 2021) such as numerical models, tracer approaches (e.g. Stable isotopes, tritium, chloride), physical approaches (e.g. Darcy's law) and remote sensing (e.g. Gravity-based approach) are routinely applied. While generally low in an arid context, recharge can be highly variable both in space and time, making its estimation even more complex (Scanlon et al., 2006).

Local-scale studies have demonstrated that whilst mainly recharged under past wet climatic conditions, the Saq-Ram Aquifer System is today receiving a modest modern recharge. A groundwater model applied on the Tabuk region of Saudi Arabia (Kawecki and Pim, 1987 in Lloyd and Pim, 1990) yielded a lateral groundwater flow of 3.1 mm yr$^{-1}$ at steady state which indicates that the natural recharge must be a minor proportion of this figure considering the
regional water table decline initiated in the mid-1980s. More recently, Al-Sagaby and Moallim (2001) applied a chloride mass-balance method on a sand dune located in the Al Qasim region and derived 1.8 mm yr$^{-1}$ of natural recharge, i.e. 2.5 % of the annual average rainfall (AAR). Other chloride mass-balance approaches were applied to small alluvial aquifers of the Asir and Hijaz mountains (along the Red Sea in western Saudi Arabia; Bazuhair and Wood, 1996)
yielding recharges equal to approximately 3.5 % of the AAR. A Darcy's law method involving the average hydraulic gradient observed on the Saq sandstone outcrops south of Tayma revealed a natural recharge of 2.5 mm yr$^{-1}$ (i.e. 4.3 % of the AAR; BRGM and Abunayyan Trading Corp., 2008). The methods used in most of these studies are relevant and convenient when applied at local scales, but much more challenging to integrate over large multi-layered aquifers such as
the Saq-Ram Aquifer System.

    Since 2002, the GRACE (Gravity Recovery And Climate Experiment) twin-satellite system has provided monthly Earth gravity measurements over large domains, variations of which are

chiefly due to mass changes in water bodies (Landerer and Swenson, 2012). GRACE gravity variations integrate terrestrial water storage variations which include groundwater storage variations over large regional-scale aquifers. Without the use of specific downscaling approaches, these selected aquifers are usually larger than $0.1 \times 10^6$ km² given the 3°×3° native spatial resolution of the GRACE data (filtered up to 1°×1° for some products; Landerer and Swenson, 2012; Wiese et al., 2016). As it is much less time- and cost-consuming than ground-based methods, this remote sensing approach has been widely used for quantifying variations in aquifer storage (e.g. Bonsor et al., 2018; Othman et al., 2018; Ramillien et al., 2014; Richey et al., 2015; Scanlon et al., 2021; Sun, 2013). Some studies implemented GRACE-derived *GWS* variations in regional-scale mass-balance equations to estimate the domain-averaged recharge for large regional aquifers of the Saharan belt (from $0.6 \times 10^6$ km² to $2 \times 10^6$ km²; Gonçalvès et al., 2013; Mohamed et al., 2017; Mohamed and Gonçalvès, 2021).

Likewise, Fallatah et al. (2019) applied this approach to the Saq Aquifer System, considering a 440 000 km² surface area domain excluding the Jordanian part (i.e. Ram Aquifer) even though it has been recognized as hydrologically connected to the Saq (UN-ESCWA and BGR, 2013). Using a unique GRACE solution (i.e. CSR RL05M v1) over the 2002-2016 period, their study yielded 11.9 mm yr$^{-1}$ of total recharge, i.e. natural plus artificial recharge (mostly irrigation return flow from agricultural practices and some wastewater reinjection in large cities). Subtracting an irrigation return flow of about 2.3 mm yr$^{-1}$ (using a plausible 15 % out of the $7\,800 \times 10^6$ m$^3$ yr$^{-1}$ mean 2002-2016 agricultural pumping; Gonçalves et al., 2013) yields a residual natural recharge estimate of 9.6 mm yr$^{-1}$, which is significantly higher than previous estimates from local studies mentioned above (i.e. approximately 2.5 mm yr$^{-1}$). This natural recharge estimated using Fallatah et al. (2019) equates to about 18 % of the AAR over the Saq Aquifer System while similar arid-zone aquifers (in the Sahara) present recharges of $(1.8 \pm 0.3)$ % of the AAR with the same GRACE-derived method (Mohamed and Gonçalvès, 2021). Furthermore, MacDonald et al. (2021) established a recent synthesis of groundwater recharge estimates in Africa using more local methods (chloride mass balance, environmental tracers, water balance, calibrated groundwater models and soil physics methods). Considering only the arid regions presenting annual precipitations below 150 mm, their study revealed on average recharges corresponding to $(3.3 \pm 5.5)$ % of the AAR.

The present study proposes a reappraisal of the GRACE-derived water budget of the combined Saq-Ram Aquifer System, accounting for groundwater withdrawal, natural drainage and artificial recharge in order to provide a long-term estimate (2002-2019) of the domain-averaged natural recharge. The three existing GRACE solutions (JPL, CSR, GSFC) were tested for their local compatibility to compute groundwater storage variations by combining with the three soil moisture datasets (VIC, CLSM, NOAH) available from the Global Land Data Assimilation System (GLDAS) land surface models. The relevance and consequences of such an application of the GRACE-GLDAS approach on arid aquifers in terms of temporal resolution of the recharge are discussed. Leading to domain-averaged values for the groundwater fluxes, the integrative approach proposed here was then refined by identifying first-order spatial heterogeneities regarding the recharge of the Saq-Ram aquifer: i) the comparison between the natural recharge obtained here and the values identified for other large arid aquifers is used to highlight a major geological control on recharge distribution; ii) distributed over the agricultural areas, the overall  irrigation excess provides a reliable estimate of the artificial recharge; iii) potential local disconnection between the groundwater recharge and the declining water table, causing a local zero recharge in areas of extensive exploitation, is discussed.

## 2. Material and methods

### 2.1 Study site

Underlying the Nafud desert of the northern Arabian Peninsula, the Saq-Ram Aquifer System (520 000 km² spanning 10° longitude and 8° latitude) constitutes the main water resource supporting vital needs of more than 3.5 million people (considering the Disi-Mudawarra water conveyance system in Jordan; UN-ESCWA and BGR, 2013). Since the mid-1980s, its intensive exploitation through large irrigated areas (Figure 1) has been quickly recognized as causing a depletion of the groundwater resource, making it one of the most overstressed aquifer systems worldwide to date (Richey et al., 2015).

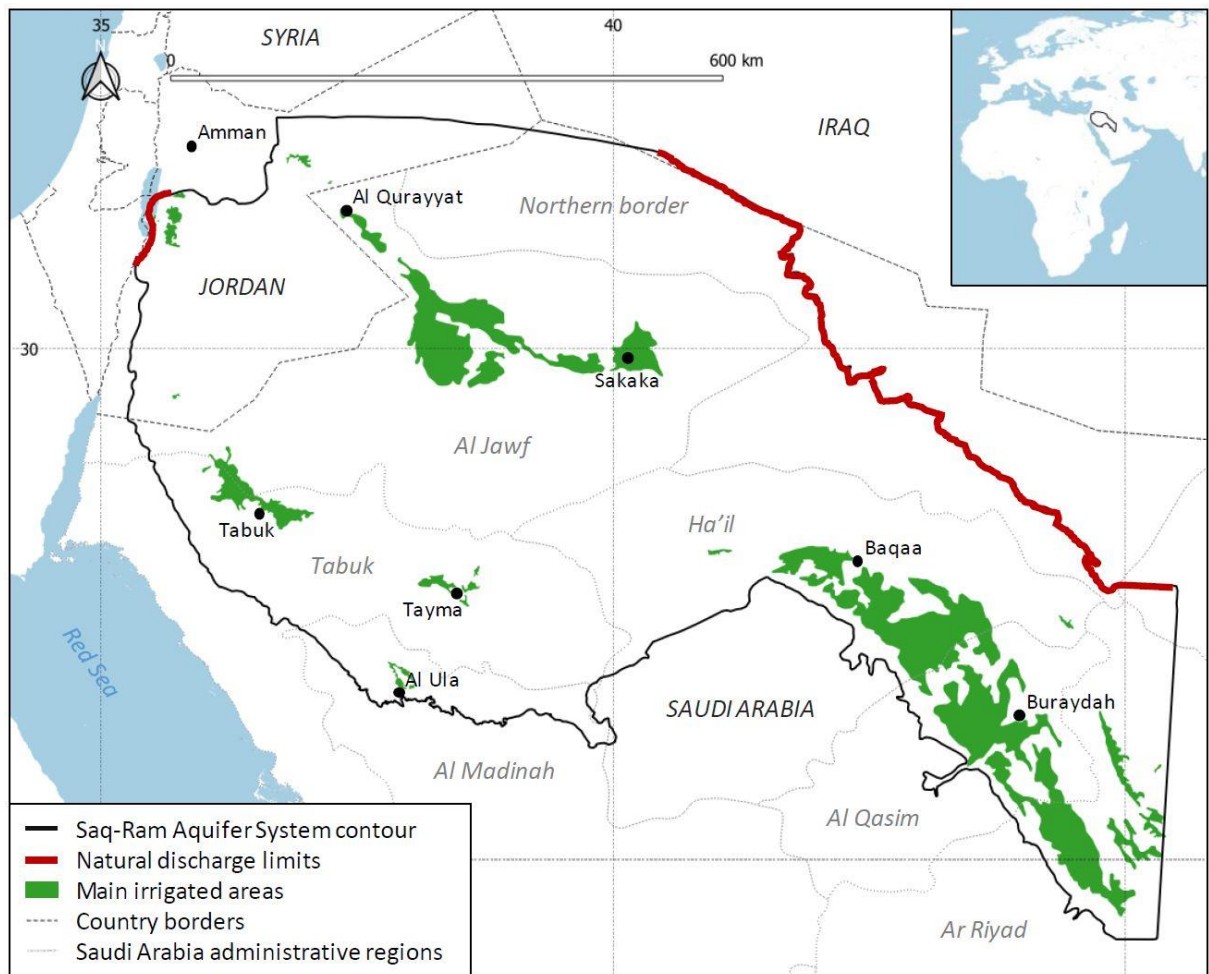

**Figure 1**: Context map of the Saq-Ram Aquifer System (WGS84 coordinates shown by straight dotted lines every 5 degree; Shorelines and country borders extracted from Wessel and Smith, 1996; Administrative regions extracted from www.gadm.org).

Mean annual temperature is about 26°C ranging between 27 and 43°C in summer, and between 8 and 20°C in winter. Typical of other arid domains, recharge (when it occurs) is associated with intermittent rainfall events (de Vries and Simmers, 2002). Extracted from the Climatic Research Unit database (CRU; Harris et al., 2020), the long term annual average rainfall (AAR) over the studied domain is 55 mm yr$^{-1}$ (over the period 1901-2019), with a maximum of 80 mm in 1982, and a minimum of about 40 mm in 1978 and 2009 (Figure 2).

## 2.2 Hydrogeological context

### 2.2.1. Definition of the reservoirs

Mainly located in the northern half of Saudi Arabia, and shared with Jordan, the overall Saq-Ram Aquifer System includes multiple sub-systems of different depths and ages (Paleozoic, Mesozoic and Cenozoic) which are mostly interconnected (see UN-ESCWA and BGR, 2013 for more details about these sub-systems):
- Paleozoic and Mesozoic deep Aquifer Systems:
    - The most productive Saq-Ram formation (West Saudi Arabia and Jordan)
    - Wasia-Biyadh-Aruma Aquifer System (North Saudi Arabia)
- Cenozoic shallow Aquifer Systems:
    - Basalt Aquifer System: Azraq-Dhuleil basin (Jordan)
    - Tawil-Quaternary Aquifer System: Wadi Sirhan Basin (North Saudi Arabia)
    - Umm er Radhuma-Dammam Aquifer System: Widyan-Salman (Saudi Arabia, Iraq, and Kuwait)
    - Other Cenozoic and Quaternary overlying formations (e.g. Harrats)

Hence, a succession of sedimentary formations (chiefly sandstone) overlies unconformably the southern Precambrian basement composed of igneous and metamorphic complexes. Dipping gradually towards the north-east to reach a 10 km thickness near the Arabian Gulf, these formations become locally confined (e.g. Hanadir and Qusaiba shale formation) but stay globally interconnected especially by the late Cenozoic reactivation of the Kahf fault systems (Othman et al., 2018). The Tabuk-Al Ula and Al Qurayyat-Al Jawf regions are characterized by overlying volcanic lava deposits called Harrats. These wide and thick heterogenous reservoirs constitute one of the largest multi-layered aquifer systems worldwide.

### 2.2.2. Natural discharge

In order to limit as much as possible the error in the water balance associated with the lateral drainage, this study focuses on the "large" Saq-Ram Aquifer System contour (Figure 1) defined by Barthélemy et al. (2006). In this large-scale domain, the contact with the crystalline basement along the southwestern margin, and the northern aquifer limit can be considered as no flow boundaries. A drainage boundary toward the eastern flank of the Dead Sea was estimated at $54 \times 10^6$ m$^3$ yr$^{-1}$ by Siebert et al. (2014), or ranging between $30 \times 10^6$ m$^3$ yr$^{-1}$ and $90 \times 10^6$ m$^3$ yr$^{-1}$ (Salameh, 1996; Lensky et al., 2005; in Siebert et al., 2014). A further $80 \times 10^6$ m$^3$ yr$^{-1}$ groundwater drainage along the Iraqi border, toward the Umm er Radhuma-Dammam Aquifer System (and ultimately the Euphrates basin) was reported in Frenken and UN-FAO (2009). Other unquantified outlets exist (e.g. drainage toward the southern basement and the Khuff aquifer at the Southeast) but these are likely minor factors in the water budget compared to other outflows in particular agricultural withdrawals. Moreover, Alsharhan and Nairn (1997) showed that the main Saq formation disappears around where the eastern vertical border (~45°E, Figure 1) has been outlined by Barthélemy et al. (2006). Finally, with regard to historical piezometric maps of the Saq aquifer (Sharaf and Hussein, 1996; Lloyd and Pim, 1990), it can be assumed that the southeastern limit with the Khuff aquifer is likely inactive given the large drawdown cone created by the intensive pumping of the Al Qasim area.

### 2.2.3. Groundwater pumping

Groundwater pumping time series can be challenging to obtain in regions where the regulation and monitoring of withdrawals was implemented only recently, as in Saudi Arabia. For countries where most of the water resource is used for agriculture, these data are often estimated using proxies such as agricultural surfaces identified by satellite imagery combined with estimates of irrigation doses supplied per crop type. BRGM and Abunayyan Trading Corp. (2008) applied such a method for each region of Saudi Arabia within the Saq aquifer (i.e. excluding the Jordanian part of the Saq-Ram Aquifer System) over the 1971-2003 period (Figure 2). Othman et al. (2018) presented a time series (1970-2015) of the cumulative pumping amounts for Al Qasim, Ha'il and Al Jawf regions (Figure 1) with data provided by the Water Resources Development Department in the Ministry of Environment, Water and Agriculture (MEWA) of Saudi Arabia (Figure 2). This updated time series revealed a good correlation ($R^2$ = 0.996) with the 1971-2003 BRGM estimates (indicating the same source of data), and the three regions considered account for the majority of agricultural withdrawals (85 % on average) from the Saq aquifer. In addition, Alhassan et al. (2016), and Chowdhury and Al-Zahrani (2013) provided regional values of the agricultural water demand for the 2009-2012 period yielding an average $7.7 \times 10^9$ m$^3$ yr$^{-1}$ for the Saq aquifer. Domestic withdrawals for 2003 are given by the BRGM and Abunayyan Trading Corp. (2008) for each region of Saudi Arabia yielding $300 \times 10^6$ m$^3$ yr$^{-1}$ for the entire Saq aquifer in that year. The 2003 industrial water demand is also reported by the same authors at about $20 \times 10^6$ m$^3$ yr$^{-1}$.

Finally, Jordan groundwater pumping data are given by region in 2015 and 2017 by the Jordan Water Sector Facts and Figures (Almomani et al., 2015; url: www.mwi.gov.jo) both reporting about $300 \times 10^6$ m$^3$ yr$^{-1}$ for the Jordanian part of the Saq-Ram Aquifer System.

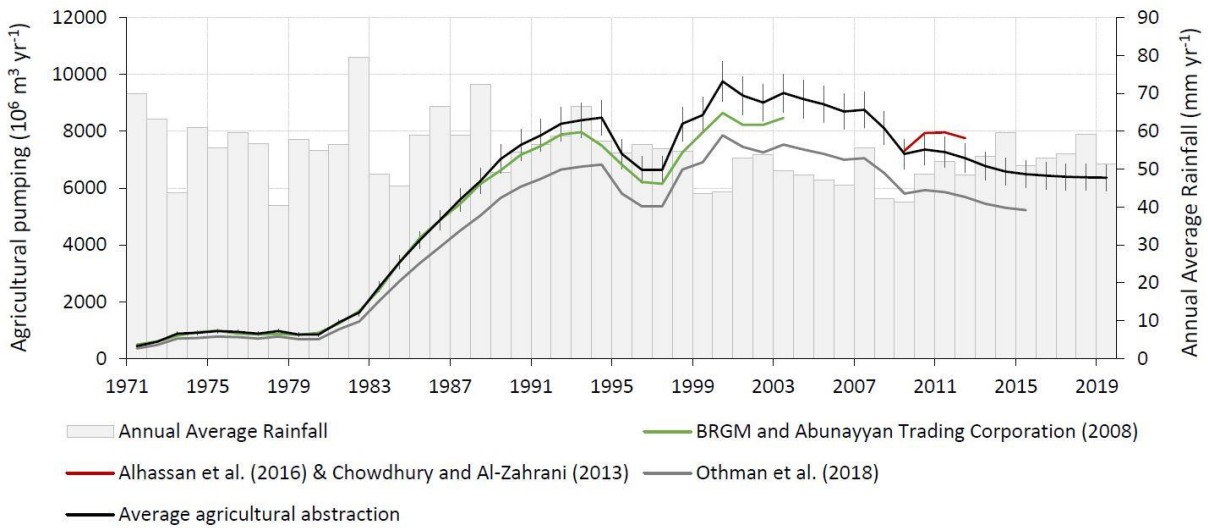

**Figure 2**: Annual average rainfall (Climatic Research Unit; mm yr$^{-1}$) of the Saq-Ram Aquifer System and agricultural withdrawal (from different sources represented by colored lines; $10^6$ m$^3$ yr$^{-1}$) of its Saudi part (except for Othman et al.'s (2018) data corresponding to Al-Qassim, Ha'il and Al-Jouf regions of Saudi Arabia). Since most of this previously published data comes from governmental entities, no associated uncertainty is provided with it.

### 2.3 Gravity and Soil Moisture data

The GRACE twin-satellite system launched in 2002 by NASA and the German Aerospace Center (DLR) monitors the Earth's gravity at a 3°×3° spatial resolution on a monthly basis. Followed by GRACE-FO (GRACE-Follow on) in 2018, the mission measures gravity

anomalies (i.e. gravity value of a given month minus the average value from January 2004 to December 2009), which are chiefly due to mass variations in water bodies (groundwater, soil water, and surface water). Monthly gravity anomalies, denoted terrestrial water storage (*TWS*), are expressed in water height (i.e. Equivalent water thickness) and can thus be used for hydrological mass-balance calculations at the regional scale. After more than one decade of

gravity field processing using spherical harmonics (Landerer and Swenson, 2012; Swenson and Wahr, 2006), an alternative interpretation using Mass Concentration blocks (termed Mascons) of discrete spherical caps at the Earth's surface was proposed (Watkins et al., 2015). The main advantage of this new interpretation is lower geophysical signal losses inducing less post-processing treatment. Provided by different computing centers (GRACE/GRACE-FO Mascon

data are available at http://grace.jpl.nasa.gov), three alternative solutions of Mascon interpretations were retrieved: (i) Jet Propulsion Laboratory (JPL; 3×3° squared tiles; Watkins et al., 2015; Wiese et al., 2019); (ii) Centre for Space Research (CSR; 1×1° hexagonal tiles; Save, 2020; Save et al., 2016); (iii) Goddard Space Flight Center (GSFC; 1×1° squared tiles; Loomis, 2020; Loomis et al., 2019).

The proposed scaling factors which correspond in fact to downscaling factors (Scanlon et al., 2016) were not used here. In fact, these downscaling factors are based on the mass distribution calculated by land surface model (LSM) accounting for surface and subsurface water transfers (Landerer and Swenson, 2012) while *TWS* variations in such arid regions are expected to be chiefly controlled by groundwater mass variations. Moreover, as stated by the

authors, the use of such gain factors is not suitable to quantify trends. We used 187 monthly observations for the period between April 2002 and July 2020. The trend of a time series can be analyzed and interpreted in terms of variation in water amounts (e.g. in mm yr$^{-1}$) and thus net water fluxes over a domain of interest.

The Global Land Data Assimilation System (GLDAS; data available at

https://ldas.gsfc.nasa.gov/gldas) provides soil water storage (*SWS*) for the first couple of meters by combining ground-based, satellite data and hydrological surface model results (Rodell et al., 2004). Matching the GRACE observation dates, we retrieved the 187-monthly data from the 2.1 versions with a 1°×1° spatial resolution computed with: (i) the Variable Infiltration Capacity model (VIC; Beaudoing et al., 2020a); (ii) the Community Land Surface Model (CLSM; Li et

al., 2020); (iii) the National Oceanic and Atmospheric administration model (NOAH; Beaudoing et al., 2020b).

The *SWS* anomalies were computed the same way as for the GRACE *TWS* anomalies, i.e. by subtracting the January 2004 to December 2009 average from each monthly value of the time series. Both the GRACE products and GLDAS solutions were spatially averaged using the

surface weight of each polygon within the Saq-Ram Aquifer domain (Figure 3).

While the time series of the three GRACE solutions (Figure 3a) clearly show a significant decrease of the *TWS* chiefly due to an increasing groundwater deficit (no permanent surface water bodies over the studied domain), the GLDAS products reveal a three-phase behavior of the *SWS* signal (Figure 3b) which can be related to variations of the climatic inputs of the

respective models: a great variability from 2002 to 2006; small variations and values close to zero, or poorly negative, from 2007 to 2018; and high variability and positive values after 2019.

The increasing discrepancies among the GRACE-JPL, -CSR and -GSFC products for *TWS*, which is clearly apparent after 2012, are mainly due to the diverse shape and size of the Mascons, and the various methods of eliminating signal leakage effects used by the three

respective computing centers. Lenczuk et al. (2020) acknowledge that the GRACE-GSFC product differs the most from the others.

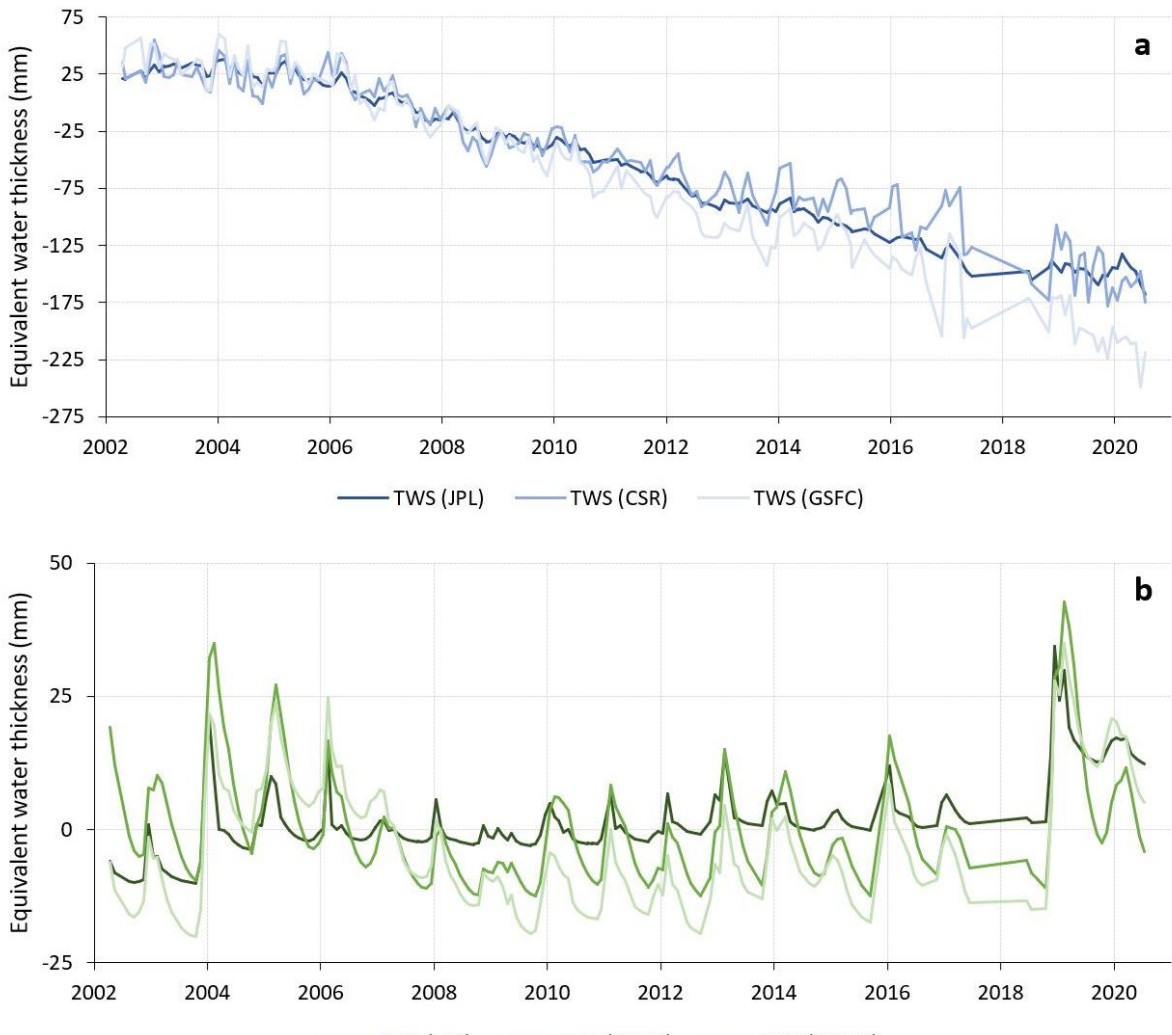

**Figure 3**: Monthly times series of (a) the GRACE-JPL, -CSR, -GSFC terrestrial water storage anomalies (TWS; mm), and (b) the GLDAS-VIC, -CLSM, -NOAH soil water storage anomalies (SWS; mm) of the Saq-Ram domain.

### 2.4 Methods

The data is used to build a regional-scale water mass-balance and estimate recharge from variations in groundwater storage (*ΔGWS*). Usually obtained by piezometric measurements, it is difficult to obtain spatially meaningful averages of *ΔGWS* over large and poorly monitored regions such as the Arabian Peninsula. GRACE data provide a valuable integrated mapping of water storage over large domains. In arid areas, surface water can generally be neglected. Moreover, the Saq-Ram aquifer is devoid of any permanent surface water bodies. Thus, using GLDAS soil water storage (*SWS*) estimates, the first step was to isolate the groundwater storage (*GWS*) anomalies from the terrestrial water storage (*TWS*) anomalies expressed in water height with the following simple decomposition of the GRACE data:

$$TWS = GWS + SWS \qquad (1)$$

And thus:

$$\Delta GWS = \Delta TWS - \Delta SWS \qquad (2)$$

Upon identification of a long-term trend (18 years in our case) of the *GWS* signal (using a $4^{th}$ order polynomial regression to filter out the seasonal signal), *ΔGWS* is calculated and can

be introduced in an overall groundwater balance equation to identify one unknown (e.g. recharge) if all other fluxes are known. The following water budget can be expressed for each of the GRACE solutions:

$$\Delta GWS = R - Q_w - Q_d \tag{3}$$

with $R$, the total recharge, $Q_w$, the groundwater withdrawal, and $Q_d$, the natural discharge.

By estimating the artificial recharge flow $R_a$ (i.e. irrigation return flow and domestic water return flow) it is possible to estimate only the natural contribution $R_n$ to the recharge of the Saq-Ram Aquifer System:

$$R_n = \Delta GWS + Q_w + Q_d - R_a \tag{4}$$

Long-term time series of groundwater withdrawal (Sect. 3.1) and artificial recharge flow (Sect. 3.2) are thus required over the GRACE period considered here (2002-2019) in order to compute the natural recharge rate. Note that the main natural discharge flows ($Q_d$) previously estimated by other studies have been already listed in the Hydrogeological context (Sect. 2.2).

## 3. Results

### 3.1 Groundwater pumping

Groundwater withdrawals from the Jordanian part of the Saq-Ram Aquifer System were estimated with the 2015 and 2017 Jordan Water Sector Facts and Figures (Almomani et al., 2015; url: www.mwi.gov.jo). Using the 2000-2017 evolution of the number of wells per usage and the 2015 and 2017 total groundwater pumping volumes per region, the 2002-2019 time series were reconstructed by extrapolating the trend of the previous years to the 2018 and 2019 missing values. This resulted in $(210 \pm 30) \times 10^6$ m$^3$ yr$^{-1}$ of mean agricultural pumping, and $(60 \pm 10) \times 10^6$ m$^3$ yr$^{-1}$ of domestic uptake for the Jordanian part of the Saq-Ram domain over the 2002-2019 period.

Using previously published data (see Sect. 2.2), we can also estimate a continuous agricultural pumping time series for the Saudi part of Saq-Ram Aquifer System (Figure 2). First, we computed linear regressions between the Othman et al. (2018) time series (grey curve in Figure 2) and (i) BRGM and Abunayyan Trading Corp. (2008) data (green curve), (ii) Alhassan et al. (2016), and Chowdhury and Al-Zahrani (2013) data (red curve) given per region of Saudi Arabia. Subsequently, as shown in Figure 2, the mean agricultural withdrawals and their uncertainties were obtained (black curve) by averaging these two regressions. In the absence of data for the 2016-2019 period, we extrapolated the trend of the previous years assuming a similar evolution of the agricultural coverage as documented in the recent Statistical Yearbook from the Ministry of Economy and Planning of Saudi Arabia (General Authority for statistics, 2019; url: www.stats.gov.sa/en/46). This yielded an average agricultural uptake of $(7\,600 \pm 500) \times 10^6$ m$^3$ yr$^{-1}$ for the Saudi part of the Saq-Ram domain over the 2002-2019 period.

Domestic pumping was reconstructed for the 2002-2019 period based on the BRGM and Abunayyan Trading Corp. (2008) domestic withdrawals per region of Saudi Arabia observed in 2003, and the Saudi Arabia population growth figures per region (United Nations, 2019). For the 2002-2019 period, we obtained an average of $(370 \pm 50) \times 10^6$ m$^3$ yr$^{-1}$.

Industrial pumping is unknown. Only the 2003 industrial water demand is reported in BRGM and Abunayyan Trading Corp. (2008) at $17 \times 10^6$ m$^3$ yr$^{-1}$. This corresponds to 0.2 % of the total uptakes and can thus be neglected, in particular when considering the uncertainty involved with the other pumping volumes.

### 3.2 Artificial recharge

The artificial recharge by irrigation return flow and wastewater reinjection was considered in this study in order to separate artificial and natural recharge. Based on a spatial decision support system, Multsch et al. (2013) carried out the assessment of the so called 'Water Footprint' (WF) of the agriculture for each region of Saudi Arabia. Using the blue WF (i.e. irrigation dose, coming entirely from groundwater in Saudi Arabia) and grey WF (i.e. irrigation in excess returning to the shallow aquifer and needed to dilute soil pollutants) computed per region by the authors, it is possible to assess a weighted average Irrigation Return Flow Coefficient (IRFC) of $(11.6 \pm 5.8)$ % for the Saq aquifer domain. As the associated uncertainties of WF flows are not given, we used half of the IRFC figure as the margin of error. Applying this coefficient to the agricultural withdrawal time series, this yielded an average irrigation return flow of $(900 \pm 450) \times 10^6$ m$^3$ yr$^{-1}$ for the Saq-Ram Aquifer System (including the Jordanian contribution estimated at $(20 \pm 10) \times 10^6$ m$^3$ yr$^{-1}$) over the 2002-2019 period.

Regarding domestic water, Chowdhury and Al-Zahrani (2015) reported an average wastewater generation coefficient of about 70 % in Saudi Arabia. This study also revealed that 38 % of these wastewaters are treated by plants. Al-Jasser's (2011) data indicate that 27 % of the treated wastewater of Riyadh city does not return to the aquifer, and that the other 73 % returns to groundwater through irrigation use and direct injection in Wadis. All combined, it can be estimated from these figures that 63 % of domestic uptake returns to the aquifer (i.e. $(270 \pm 30) \times 10^6$ m$^3$ yr$^{-1}$ for the Saq-Ram Aquifer System over the 2002-2019 time period).

### 3.3 Local compatibility between GLDAS models and GRACE products

The high frequency component of the *TWS* signal corresponds to the seasonality (including soil moisture variations) which is theoretically simulated by the GLDAS Land Surface models (Figure 3). Even if only long-term trends are interpreted here, it is crucial to verify the compatibility of each of the GLDAS models with each of the GRACE solutions before assessing the *ΔGWS* (Gonçalvès et al., 2013; Scanlon et al., 2019). A soil moisture amplitude of a GLDAS model greater than the seasonality as shown by a GRACE product disqualifies this soil model for the groundwater mass balance analysis. Hence, the six time-series were detrended and their annual amplitudes compared (Figure 4). This analysis revealed that, for the Saq-Ram Aquifer System, the GLDAS-CLSM and -NOAH products are not compatible with the GRACE-JPL solution since their mean amplitudes are greater (i.e. overestimated simulated seasonality in *SWS* compared to the *TWS* observations). The other two GRACE solutions (CSR and GSFC) are compatible with the three GLDAS products with similar performance.

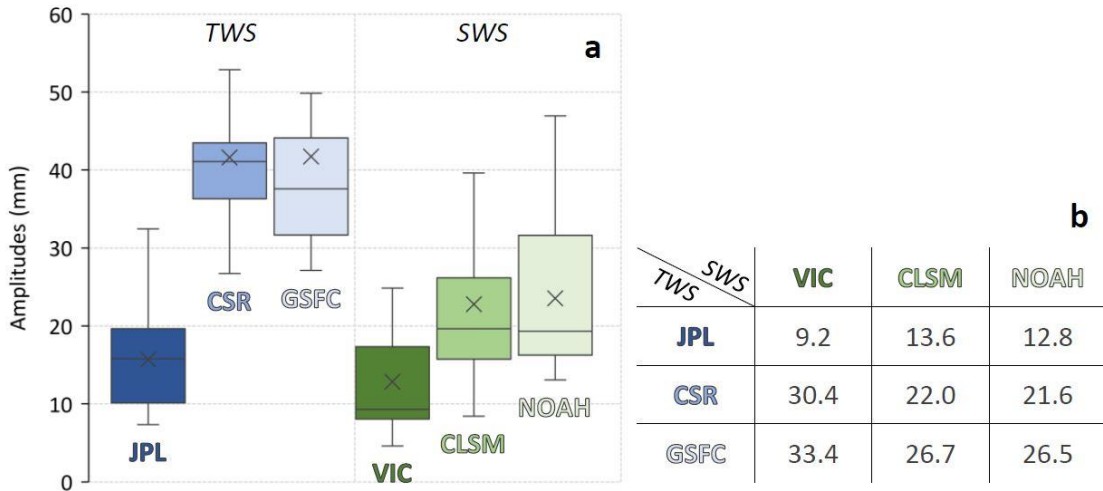

**Figure 4**: (a) Annual amplitude comparison between the three GRACE solutions and the three GLDAS products and (b) the associated RMSE of the linear regressions applied on these amplitudes.

### 3.4 GRACE-GLDAS water budgets for the 2002-2019 period

After computing the long-term average (18 years) groundwater storage variation for each of the GRACE-GLDAS solutions and implementing the fluxes previously estimated, the water budgets were calculated (Table 1).

| Water Budget | JPL Solution | | CSR Solution | | GSFC Solution | |
|---|---|---|---|---|---|---|
| **GRACE data ($TWS$)** | JPL RL06M v2 CRI | | CSR RL06M all cor. v2 | | GSFC RL06 v1 OBP-ICE6GD | |
| **GLDAS data ($SWS$)** | v2.1 VIC | | v2.1 VIC, CLSM, NOAH | | v2.1 VIC, CLSM, NOAH | |
| **Water Budget** | Mean | SD (1σ) | Mean | SD (1σ) | Mean | SD (1σ) |
| $\Delta GWS$ | -11.35 (-5 940) | 0.35 (180) | -11.18 (-5 850) | 0.52 (270) | -14.80 (-7 750) | 0.50 (260) |
| Natural discharge ($Q_d$) | 0.27 (140) | 0.06 (30) | 0.27 (140) | 0.06 (30) | 0.27 (140) | 0.06 (30) |
| Withdrawal ($Q_w$) | 15.66 (8 200) | 1.05 (550) | 15.66 (8 200) | 1.05 (550) | 15.66 (8 200) | 1.05 (550) |
| Artificial Recharge ($R_a$) | 2.23 (1 170) | 0.84 (440) | 2.23 (1 170) | 0.84 (440) | 2.23 (1 170) | 0.84 (440) |
| Natural Recharge ($R_n$) | **2.35 (1 230)** | **1.39 (730)** | **2.51 (1 310)** | **1.44 (750)** | **-1.10 (-580)** | **1.43 (750)** |
| Total Recharge ($R$) | 4.58 (2 400) | 1.62 (850) | 4.74 (2 480) | 1.67 (870) | 1.13 (590) | 1.66 (870) |

**Table 1**: Domain averaged groundwater budget in mm yr⁻¹ (10⁶ m³ yr⁻¹) of the Saq-Ram Aquifer System (520 000 km²) for each of the GRACE-GLDAS solutions (and associated standard deviation SD one sigma) for the 2002-2019 period.

The negative natural recharge, i.e. evaporation losses from the water table, obtained with the GRACE-GSFC solution is not realistic (see the following discussion; Sect. 4.1). Therefore, only the results obtained from the GRACE-JPL and -CSR solutions were considered. Hence,

the 2002-2019 average natural recharge of the Saq-Ram Aquifer System is estimated at $(2.4 \pm 1.4)$ mm yr$^{-1}$. Using the long-term annual average rainfall of 55 mm yr$^{-1}$, the domain-average natural recharge corresponds to $(4.4 \pm 2.6)$ % of the AAR.

## 4. Discussion

This study provides an estimate of the 2002-2019 domain-average natural recharge with associated standard deviation accounting for temporal variations in natural discharge, groundwater pumping and irrigation return flow. The uncertainties associated with the calculation of the $\varDelta GWS$ long-term trends with the GRACE and GLDAS products have also been considered. Errors associated with GRACE measurements could not be accounted for as they are not provided with the raw Mascons data (i.e. before the application of unwanted scaling factors). However, Blazquez et al. (2018) investigated the uncertainty of raw GRACE data by solving a global water budget using trends in ocean mass, ice loss from Antarctica, Greenland, arctic islands and trends in water storage over land and glaciers. The authors estimated a 0.27 mm yr$^{-1}$ uncertainty for the GRACE data, a figure significantly lower than the uncertainties of the $\varDelta GWS$ trends used in this study (Table 1).

Even if the Saq-Ram domain is devoid of any permanent surface water bodies, ephemeral streams are known to be important for (eco)hydrology and local groundwater recharge in arid regions (Shanafield et al., 2021; Dogramaci et al., 2015; Schilling et al., 2021). However, runoff coefficients were estimated at about 1% in the region (Al-Hasan and Mattar, 2013) while more than 90% of this runoff is lost by evaporation in the low lands. Thus, accounting for recharge redistribution through ephemeral streams in the domain-average water budget of the large Saq-Ram aquifer system would be quantitatively insignificant.

### 4.1 Negative natural recharge of the GRACE-GSFC solution results

The GRACE-GSFC solution resulting in negative natural recharge suggests that the Saq-Ram Aquifer System is subject on average to 1.1 mm yr$^{-1}$ of evaporation losses from the water table. A clear relationship between evaporation losses (also called evaporative pumping) and vadose zone thickness has long been demonstrated (Coudrain-Ribstein et al., 1998; Fontes et al., 1986; Kamai and Assouline, 2018; Zammouri, 2001). Stable isotope measurements by Fontes et al. (1986) in northern Sahara revealed that a groundwater evaporation rate of 2.0 mm yr$^{-1}$ is reached for an average 10 m vadose zone thickness (between 6 and 30 m for 1.0 mm yr$^{-1}$ of groundwater evaporation rate according to Coudrain-Ribstein et al., 1998). Two recent studies (Ahmed et al., 2015; Zaidi et al., 2015) computed a mean vadose zone thickness of about 150 m (ranging from 15 to 300 m) for a domain including the vast majority of the Saq-Ram Aquifer System. According to Kamai and Assouline (2018), a vadose zone thickness of 150 m would induce a mean groundwater evaporation loss of about 0.07 mm yr$^{-1}$, well below 0.2 mm yr$^{-1}$. This discrepancy suggests that the GRACE-GSFC solution is not suitable for this study (probably due to differences in the treatment of the raw GRACE data with the other two products), and that its results may not yield a meaningful mass balance. We therefore interpreted only the JPL-CSR solutions that yield positive water balances (Table 1).

### 4.2 Contribution of the volcanic lava deposits (Harrats) to the recharge

At 55 mm yr$^{-1}$ of long-term average rainfall, the natural recharge of the Saq-Ram Aquifer domain corresponds to $(4.4 \pm 2.6)$ % of incoming precipitation. This value can be compared to

the average (1.7 ± 1.2) % recharge-AAR ratio obtained by similar gravity-based approaches on Saharan aquifers with similar hydrogeological characteristics: Murzuq Basin in Libya (0.6 × $10^6$ km²; Mohamed and Gonçalvès, 2021 using GRACE data from Bonsor et al., 2018); Nubian Sandstones System (NSAS) covering Egypt, Libya, Tchad and Soudan (2.1 × $10^6$ km²; Mohamed et al., 2017); North Western Sahara Aquifer System (NWSAS) in Algeria, Tunisia and Libya (1.2 × $10^6$ km²; Mohamed and Gonçalvès, 2021); Tindouf basin in Algeria and Morocco (0.3 × $10^6$ km²; Gonçalvès et al., Submitted); and Djeffara basin in Tunisia and Libya (0.1 × $10^6$ km²; Gonçalvès et al., 2021). A clear positive offset compared to the regression is observed for the Saq-Ram Aquifer System (Figure 5). One of the main differences between the Saq-Ram and the five Saharan basins is that the latter correspond to almost purely sedimentary porous domains, while the Saq-Ram presents a substantial proportion of overlying volcanic lava deposits called Harrats (8 % of the domain, i.e. 40 000 km² represented by black areas in Figure 5), over which a higher local precipitation is observed (i.e. AAR of 60 mm $yr^{-1}$ using CRU data from Harris et al., 2020).

To our knowledge, there is no study reporting recharge rates on these basaltic deposits strictly within the studied domain, but data are available for other parts of the Harrat Al-Sham (Syria, Jordan, Saudi Arabia; also called Al Harrah in Saudi Arabia) which corresponds to the largest volcanic lava deposit of the Arabian Peninsula, partly extending over the Saq-Ram Aquifer System. Dafny et al. (2003) reported that the recharge of the northern Golan heights (located at the western extremity of the Harrat Al-Sham) corresponds to 30 % of AAR, consistent with the piezometric contour map revealing a major recharge area in this thick volcanic formation. For the Amman-Zarqa Basin (Jordan) located more in the central part of the Harrat Al-Sham, Mahamid (2005) derived a recharge of about 18 % of AAR through a steady state model applied on the upper layer which includes a large proportion of limestone formation (B2/A7) underlying the basalt formation (forming a multi-layer aquifer). Provided as supplementary material, we applied a Water Table Fluctuation method on the average seasonal piezometric signal of the limestone/basaltic formation of the very same Amman-Zarqa Basin documented by Al-Zyoud (2012) and Al-Zyoud et al. (2015) yielding a recharge of about (29 ± 3) % of AAR. Other areas of high recharge rate within the Harrat Al-Sham are revealed by the piezometric heights reported in different studies: the Jebel Al-Druze volcanic cone (Syria; Al-Homoud et al., 1995), as well as a basaltic relief located between the cities of Safawi and Ruwaished (Jordan; Abu-Jaber et al., 1998) which is partly in the Saq-Ram domain. Moreover, another major overlying volcanic deposit, i.e. Harrat Al-'Uwayrid (located between the cities of Tabuk and Al-Ula), corresponds to a preferential recharge area for the Saq-Ram formation as pictured by a water table mound observed on the contour map of Lloyd and Pim (1990).

Considering the three values previously mentioned leads to an average recharge-AAR ratio of (26 ± 6) % for the Harrat Al-Sham. Recharge-AAR ratios about ten times those of porous sedimentary basins (1 to 3 %) have also been noted for karstic aquifers in arid environments (Gonçalvès et al., 2021; Messerschmid and Aliewi, 2021). Basalts and karst formations share common characteristics which may explain a 'funnel' role for precipitations: intense fracturing, thin soil and scarce vegetation.

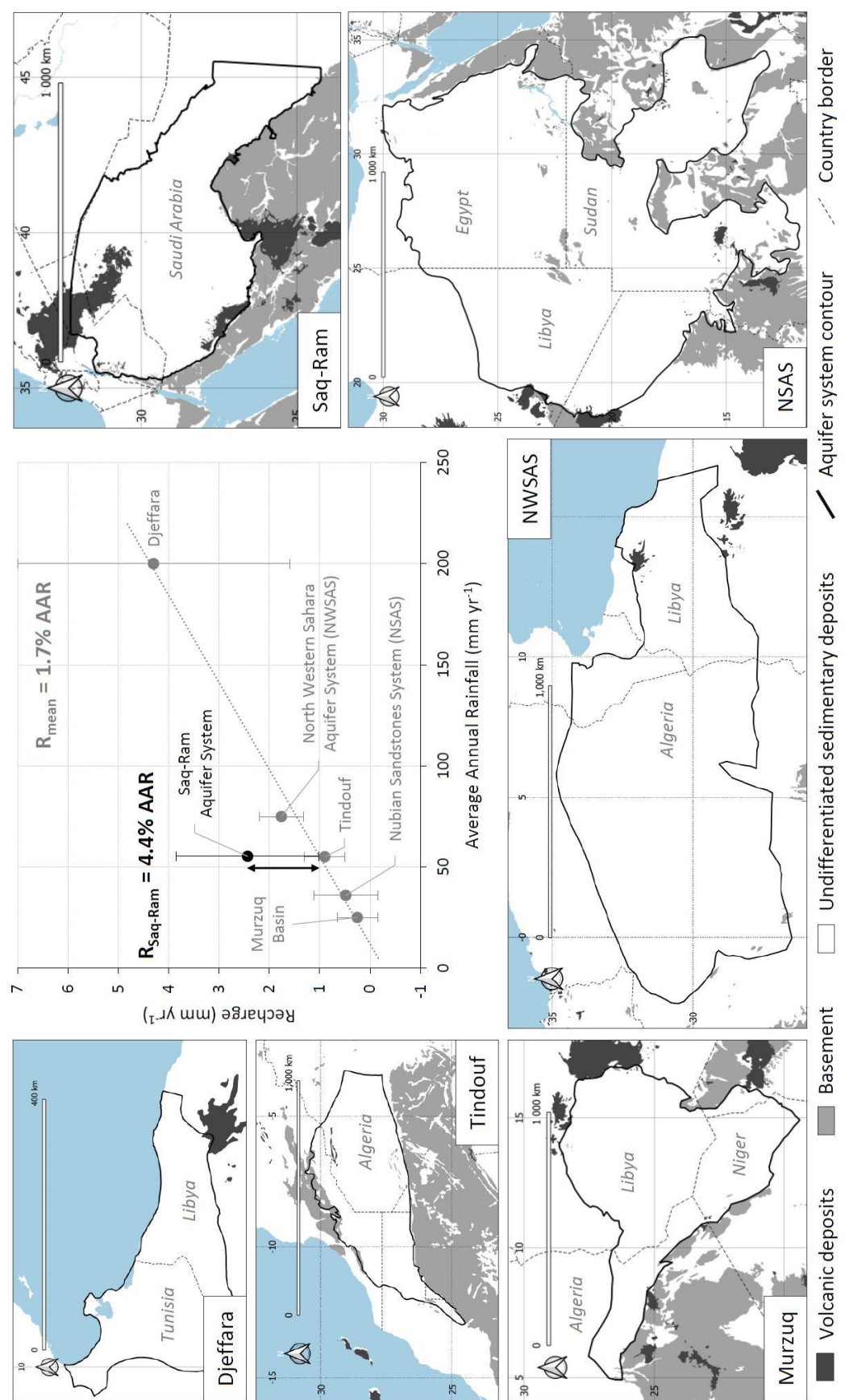

**Figure 5**: Recharge (R) versus Annual Average Rainfall (AAR) for the Saq-Ram Aquifer System, compared to other Sahara aquifer systems assessed with a similar approach, and their associated simplified geological maps (Pollastro et al., 1999; Persits et al., 1997).

510

Hence, considering average values of $(26 \pm 6)$ % and $(1.7 \pm 1.2)$ % of recharge-AAR ratios over the volcanic and porous sedimentary domains respectively, we can assess the average recharge of the Saq-Ram Aquifer System independently from any data previously used in this study. This yields a total natural recharge of $(2.1 \pm 0.7)$ mm yr$^{-1}$, consistent with the results obtained using the GRACE-GLDAS water budget approach (i.e. $(2.4 \pm 1.4)$ mm yr$^{-1}$), with a $(0.9 \pm 0.6)$ mm yr$^{-1}$ contribution to the total domain-averaged recharge by the porous sedimentary outcrops and a $(1.2 \pm 0.3)$ mm yr$^{-1}$ of contribution by volcanic lava deposits. In other words, these volcanic lava deposits (Harrats), which cover 8 % of the Saq-Ram Aquifer System, contribute to more than 50 % of the total natural recharge.

### 4.3 Local recharge velocity and water table decline

Using the Al-Sagaby and Moallim (2001) study, it is possible to estimate the recharge velocity through a sand dune located in the Al Qasim region (within the Saq-Ram domain). An average natural recharge of 1.8 mm yr$^{-1}$ obtained by chloride mass-balance together with a mean measured water content on the vadose zone of 0.01% yields a local pore velocity equivalent to a 'natural recharge front velocity' of about 0.2 m yr$^{-1}$.

It is interesting to compare this recharge velocity with the water table decline velocity. By definition, this sand dune area is located away from any agricultural plot (i.e. zero artificial recharge by irrigation return flow) but within one of the largest drawdown areas worldwide (about 500 km diameter; Sharaf and Hussein, 1996) caused by intensive pumping. Considering a conservative 30 m water table decline in 45 years (BRGM and Abunayyan Trading Corp., 2008), a minimum 0.7 m yr$^{-1}$ decline is computed on the outskirts of this piezometric depression. This is significantly faster than the local natural recharge velocity of 0.2 m yr$^{-1}$, suggesting that the unsaturated zone is thickening faster than the percolation flows into it. Note that we estimated $(900 \pm 450) \times 10^6$ m$^3$ yr$^{-1}$ of irrigation return flow (2002-2019 average; see Sect. 3.2) corresponding to $(167 \pm 83)$ mm yr$^{-1}$ distributed only over the irrigated areas of the aquifer (i.e. about 5 400 km² for the 2002-2019 period; General Authority for statistics, 2019). Such a recharge value, two orders of magnitude larger than its natural counterpart, certainly prevents the disconnection between the recharge front and the free groundwater table in irrigated areas.

Hence, while irrigation excess is great enough to artificially sustain the recharge of the aquifer within agricultural plots, the effective recharge becomes locally and temporally zero on the outskirts of such crop areas, similar to observations of a semiarid aquifer of the North China Plain by Cao et al. (2016). Some regions behave as preferential recharge areas for the Saq-Ram Aquifer System, but a mechanism of a relative disconnection of the infiltration front with the declining water table likely occurs in intensively exploited regions, most probably in the vicinity of the main irrigated areas (represented by green areas in Figure 1) where there is no artificial recharge but still a piezometric drawdown induced by intensive pumping.

### 4.4 Influence of the vadose zone and recharge mechanism on GRACE-GLDAS interpretations in arid domains

The main characteristic of aquifers in arid domains is the presence of a thick unsaturated zone (deep water table) causing considerable lag times for ground surface-water table exchanges (Figure 6). The lag time ($n$ years) between surface infiltration ($I$) and recharge ($R$) corresponds to the transit time of water across the vadose zone from the soil surface to the water table at depth. This lag time is highly dependent on the recharge mechanism, with large lags of hundreds of years for diffuse recharge and at most one year for focused recharge (Scanlon et

al., 2006). Diffuse recharge occurs fairly uniformly over large areas (precipitation or irrigation) while focused recharge refers to a mechanism of concentration of the recharge from surface topographic depressions (e.g. perennial and non-perennial streams, lakes and playas).

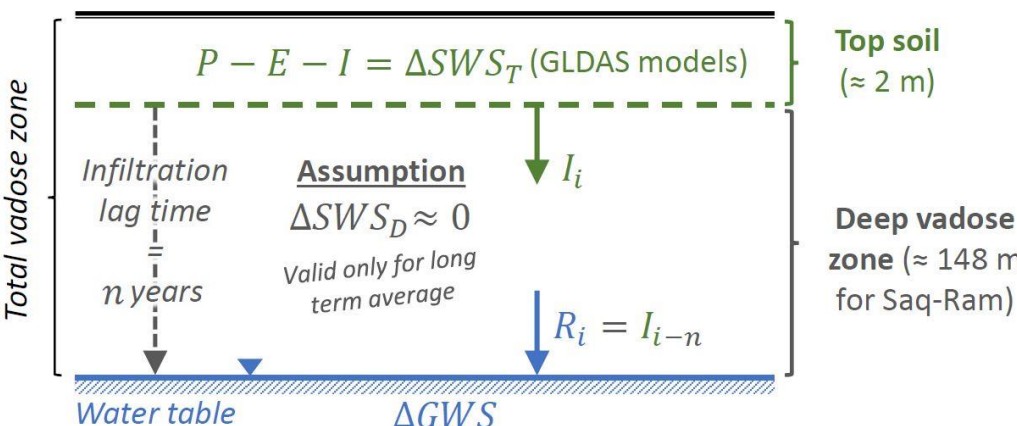

**Figure 6**: Conceptual representation of the treatment of the vadose zone in the GRACE-
GLDAS water budget approach.

    The standard interpretation of GRACE data consists in subtracting $\Delta SWS$ given by GLDAS models from $\Delta TWS$ obtained from satellite Earth gravity monitoring. However, the water balance calculated by Land Surface models in GLDAS is restricted to the top soil (about the
first 2 m, $\Delta SWS_T$; Figure 6). Therefore, the implicit assumption behind the standard interpretation is that the soil water moisture variations in the deep vadose zone ($\Delta SWS_D$ from 2 m depth to the water table elevation) are negligible which means that infiltration at 2 m from the top of this deep vadose zone ($I$) equals the recharge outflowing at its bottom ($R$). For small lag times (focused recharge), or shallow aquifers, this assumption is valid both on an annual
basis and for long-term averages. In the case of large lag times (diffuse recharge) as for the vast majority of arid domains (including the Saq-Ram Aquifer System), this assumption may be valid for long term average values ($I = R$) but not on a yearly time scale ($I_i \neq R_i$ for a specific year $i$, Figure 6). In this case, the application of a yearly GRACE-GLDAS approach will provide annual percolation rates at the bottom of the Land Surface Models ($I_i$), i.e. future
recharges ($R_{i+n}$ reaching the water table only after $n$ years of lag time). Hence, yearly analysis using GRACE-GLDAS solutions in arid domains should be restricted to areas where focused recharge is the main mechanism, while a long-term analysis is valid irrespective of the recharge mechanism (focused or diffuse).

    Regarding long-term analysis, the length of the time series considered has a significant
impact on the calculated interannual recharge (Figure 7). The computed recharge of the Saq-Ram Aquifer System and the NWSAS (Gonçalvès et al., 2013; Mohamed and Gonçalvès, 2021) is considerably different when considering different time periods of GRACE data. Compared to the NWSAS real estimate for the long-term average recharge obtained using a $^{14}$C interpretation (i.e. $(1.6 \pm 2.3)$ mm yr$^{-1}$; Chekireb et al., 2021), it appears that about 15 years of
GRACE data are required to obtain the long-term average recharge. By fitting the best threshold regression models (dashed lines), it is possible to estimate that the long-term natural recharge of the Saq-Ram Aquifer System is 2.6 mm yr$^{-1}$.

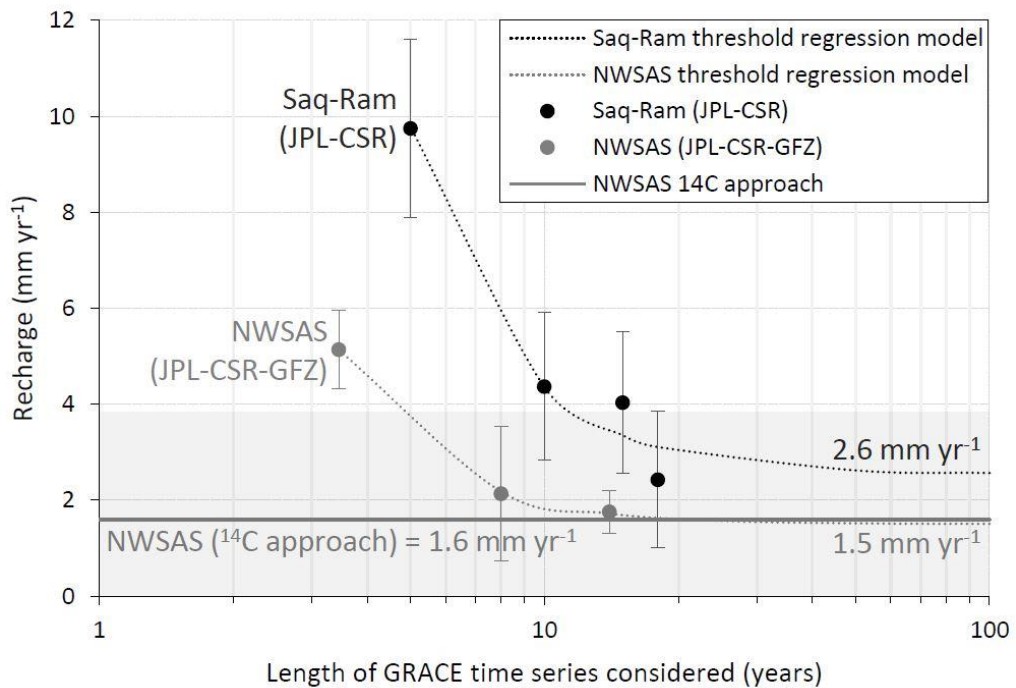

**Figure 7**: Changes in recharge rate and associated error bars considering different lengths of GRACE time series for the Saq-Ram Aquifer System compared to the NWSAS recharge rates obtained by GRACE and [14]C approaches ([14]C uncertainty as a gray square), and their associated threshold regression models.

## 5. Conclusion

In this study, we retrieved gravity data from the GRACE satellite system and soil moisture variations inferred from GLDAS models to construct a water mass balance, with the aim of estimating the long-term average natural recharge over the Saq-Ram Aquifer domain. As recommended by Gonçalvès et al. (2013) and Scanlon et al. (2019), the local compatibility of the three existing GRACE solutions (JPL, CSR, GSFC) with the soil moisture datasets available from the three GLDAS models (VIC, CLSM, NOAH) was tested to compute groundwater storage variations. Accounting for $(15.7 \pm 1.1)$ mm yr$^{-1}$ of groundwater pumping, $(2.2 \pm 0.8)$ mm yr$^{-1}$ of artificial recharge, and $(0.3 \pm 0.06)$ mm yr$^{-1}$ of natural discharge derived from previous studies, the GRACE-derived water budget yielded a $(2.4 \pm 1.4)$ mm yr$^{-1}$ of domain-averaged natural recharge over the 2002-2019 period, corresponding to $(4.4 \pm 2.6)$ % of the annual average rainfall.

In line with many other studies (local or over similar arid aquifers), we suggest that this recharge rate is spatially very heterogeneous. Volcanic lava deposits, which cover 8 % of the Saq-Ram Aquifer domain, contribute to more than 50 % of the total natural recharge when considering previously published recharge rates over the terrain. Further, due to intensive groundwater withdrawal in the last decades, a mechanism of relative disconnection from the infiltration front with the water table position (i.e. the unsaturated zone thickens faster than percolation flows through it) is suggested to occur at the outskirts of major irrigated areas, presumably making recharge locally null.

Hence, in addition to this application on the Arabian Peninsula, this study strongly indicates a major control of geological context on arid aquifer recharge which has been poorly discussed hitherto. Regarding water resource management, this work has (i) local implications: promoting more hydrogeological studies on these productive basaltic formations overlying the Saq-Ram

Aquifer System; and (ii) regional implications: questioning the impact on sustainability
calculations for numerous MENA (Middle East and North Africa) countries presenting basaltic
and/or karstic aquifers.

## Acknowledgments


We thank the *Agence Française pour le développement d'AL-ULA* (AFALULA) and the
Royal Commission of Al-Ula (RCU) for their funding of the project 'Past, present and future
Water resources in Al-Ula Oasis' (WAO) which this study is part of. The three anonymous
reviewers and the Editors are acknowledged for their relevant comments which largely
improved the manuscript.

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
