# Peer review of "Influence of intensive agriculture and geological heterogeneity on the recharge of an arid aquifer system (Saq-Ram, Arabian Peninsula) inferred from GRACE data."

_EGUsphere, 2022_

## Referee Comment (RC1)

**Review of the manuscript egusphere-2022-22**
**"Influence of intensive agriculture and geological heterogeneity on the recharge of an arid aquifer system (Saq-Ram, Arabian Peninsula)" by Seraphin P, Gonçalvès J, Hamelin B, Stieglitz T, Deschamps P**

May 8, 2022

**1 General comments**

This manuscript describes an interesting analysis of GRACE data to estimate the components of the water budget for the Saq-Ram Aquifer System.

I appreciated the careful analysis of data in order to assess which are the best products of satellite data processing to perform a water budget analysis for a wide, moderately or poorly gauged, basin. I think this is the most important and innovative aspect of the work. The remarks on effects of intensive agriculture and geological heterogeneity are also interesting, but, in my opinion, they are not so deeply considered in the work as to make them the central focus of the title.

The scientific quality of the work is good, but I would suggest to provide further discussion of aspects related to scaling, which is somehow hidden in the work-flow.

The manuscript is generally well written and well organized. Some minor technical comments are listed below.

I think that this manuscript deserves publication on a public repository. With some corrections and/or integration, it could be submitted to a high-quality international scientific journal.

**2 Specific comments**

1. Lines 87 to 97. The resolution which can be reached with the remote sensing approach should be discussed. Also the extension of the basin which can be properly studied with satellite data should be discussed. For instance, at line 96, it would be useful to explicitly state how large are the mentioned regional aquifers.

2. Line 214. The averaged water demand for agricultural purposes is given with three or four significant digits. I do not think this is physically acceptable, also taking into account the estimates of uncertainties which are explicitly given later for similar quantities.

3. Figure 3. These data deserves further comments. Any remarks about the increasing discrepancies among the products by JPL, CSR and GSFC for TWS, which is clearly apparent after 2012? Which regression is applied to TWS time series? The linear trends for SWS are not very appropriate, as these time series show, at least, a three-phase behavior: great variability from 2002 to 2006; small variations and values close to zero or poorly negative from 2007 to 2018; high variability and positive values after 2019.

4. Sections 4.3 & 4.4. I appreciate this discussion. However, the two sections could be merged in a better way. At the beginning of section 4.3 it is stated that the transit time of water through the vadose zone is about 350 years; taking into account porosity and saturation, the recharge times should be even longer. This immediately rises the concern about the fact that the dynamics of the unsaturated zone and of the aquifer recharge seems to be much slower than the variability of the gravity field. Furthermore, the time series considered here are 20 years long, which is a time interval much shorter than 350 years. At line 541, it is stated that 12 years of GRACE data are required to obtain the long-term average recharge. This seems to be in contrast with the previous comments, doesn't it?

**3 Technical comments**

1. I recommend to follow as carefully as possible the guidelines by BIPM about the use of SI units and the format with which values of physical quantities are written. In particular, I recommend to use parentheses to avoid confusion. I list here some examples.

(a) A space should be left between the value and the percentage symbol, e.g., 1 %.

(b) Line 81. "3-4%" should be substituted, possibly with "approximately 3.5 %". Similarly, at line 107.

(c) Line 109. "$1.8 \pm 0.3\%$" should be substituted with "$(1.8 \pm 0.3)$ %". Similarly, at lines 115, 345, 388, 418, 419, 445, 454, 459, 562.

(d) Lines 187 & 188. "30 and $90 \times 10^6 \mathrm{m}^3\mathrm{yr}^{-1}$" should be substituted, possibly with "$30 \times 10^6 \, \mathrm{m}^3 \cdot \mathrm{yr}^{-1}$ and $90 \, \mathrm{m}^3 \cdot \mathrm{yr}^{-1}$". Similarly at line 220.

(e) Line 312. "$210 \pm 30 \times 10^6 \mathrm{m}^3\mathrm{yr}^{-1}$" should be substituted with "$(210 \pm 30) \times 10^6 \, \mathrm{m}^3 \cdot \mathrm{yr}^{-1}$". Similarly at lines 325, 330, 355, 387, 462, 463, 465, 488, 489, 541, 559 to 561.

2. Line 51. Although Bierkens and Wada (2019) is a review paper, it could be fair to add citations of some of the seminal papers on this subject.

3. Line 54. Since the resolution of GRACE data and of the products used in this work are given in degrees, it would be useful to include the extension of the basin in latitude and longitude. This can be done here or in section 2.1.

4. Lines 151 & 152. A verb is missing in the sentence "The Climatic Research Unit... (over the period 1901-2019)", isn't it?

5. Lines 212, 318; legend of figure 2. "et al." should be substituted with "and McCluskey".

6. Figure 2. Why only one time series is shown with error bars? How is the uncertainty estimated?

7. Lines 231 & 232. Expression "its 2004.0-2009.9 average value" should be substituted, possibly with "its average value from January 2004 to September 2009"

8. Figure 3. For both plots, I could not find an explicit definition of the reference equivalent water thickness, which corresponds to a null value.

9. Lines 264 & 265. "(GLDAS data are available at https://ldas.gsfc.nasa.gov/gldas)" should be moved before, possibly at the end of the first sentence of this paragraph.

10. Line 290. Which polynomial fit is used? I mean, second, third, or higher degree? I do not see $\Delta GWS$ in Figure 3.

11. Line 343. In which sense "needed"?

---

## Referee Comment (RC3)

**Influence of intensive agriculture and geological heterogeneity on the recharge of an arid aquifer system (Saq-Ram, Arabian Peninsula)**

The paper "*Influence of intensive agriculture and geological heterogeneity on the recharge of an arid aquifer system (Saq-Ram, Arabian Peninsula)*" presents a water budget estimation of regional groundwater recharge flux for an arid aquifer system. The approach is well driven and the paper is really well written. I do have three remarks.

1) The authors claim that there is a strong spatial heterogeneity of groundwater recharge over the domain. However, such heterogeneity is barely discussed and not illustrated. A representation of the variation of the estimated recharge flux over the studied region would be a great asset. It would be interesting to have such a map and to discuss about the heterogeneity of the estimated recharge in the results. Moreover, it is not clear for now how the water budget has been computed and how the other component of the water budget has been assimilated and combined with the satellite-based products. Are the authors computed the water budget on a regular grid over the whole region? The method section (2.4) is more a theory section and is missing information about how the water budget was practically computed.

2) The authors present an estimation of the regional groundwater recharge about 2.4 mm.y$^{-1}$ (with an uncertainty of 1.4 mm.y$^{-1}$). Beside that, it is unclear if all the uncertainties presented in the paper are for 1 or 2 σ. Knowing that each component of the water budget contained large errors, it is not clear to me how the authors can have such a precise estimate of groundwater recharge. I would rather think that estimation of such a small recharge flux would be very challenging as the cumulative effects of the errors in each water budget components are also large. It would be great to have a discussion about what is really quantified in the "uncertainties" and what are the major limits of such estimation of recharge flux with a large-scale water balance approach. Also, what are the effects of boundary conditions (lines 187, 188) in the calculation of natural discharge?

3) In section 4.4, the authors are discussing the focused groundwater recharge. I do miss a complement discussion about the effects/importance of the ephemeral stream (so-called wadi systems) in groundwater recharge processes at this scale. Are these systems quantified in the approach or ignored. In both cases, it is not clear.

Minor comments

**In the abstract, I think the JPL, CSR, GSFC, VIC, etc. might be removed to facilitate the comprehension.**

\# The authors use the word "global" (e.g. line 31, 75 etc.) several times in the text. Is the word "large-scale" or "regional" would better fit? This study is not at a global scale.

To sum-up, the present paper is an interesting study lacking of some clarity in the approach and discussion about the reliability of the proposed estimates of groundwater recharge. The scientific quality is good but the scientific significance is not as well because the authors are using a well-established method and I do not see the real contribution to scientific progress for such a study. Maybe the authors can be better explicit about the real significance of such a study. The figures are good quality overall.

This manuscript can be accepted with some minor revisions in my opinion.

---

## Author Response (AR1)

Note that the **line numbers** are **based** on the **"revised_marked.pdf"** version of the manuscript (line numbers changed with Microsoft Word's revision tracking mode).
- *Reviewer comment in italic font*
- Answer to reviewer comment in regular font
- Modification of the manuscript in red

**RC1 - Anonymous Referee #1 - May 8, 2022**

**Review of the manuscript egusphere-2022-22**
"Influence of intensive agriculture and geological heterogeneity on the recharge of an arid aquifer system (Saq-Ram, Arabian Peninsula)"
by Seraphin P, Gonçalvès J, Hamelin B, Stieglitz T, Deschamps P
May 8, 2022

**1 General comments**

*This manuscript describes an interesting analysis of GRACE data to estimate the components of the water budget for the Saq-Ram Aquifer System.*
*I appreciated the careful analysis of data in order to assess which are the best products of satellite data processing to perform a water budget analysis for a wide, moderately or poorly gauged, basin. I think this is the most important and innovative aspect of the work. The remarks on effects of intensive agriculture and geological heterogeneity are also interesting, but, in my opinion, they are not so deeply considered in the work as to make them the central focus of the title.*
*The scientific quality of the work is good, but I would suggest to provide further discussion of aspects related to scaling, which is somehow hidden in the work-flow. The manuscript is generally well written and well organized. Some minor technical comments are listed below. I think that this manuscript deserves publication on a public repository. With some corrections and/or integration, it could be submitted to a high-quality international scientific journal.*

We agree with Reviewer #1 that data analysis and the choice of satellite products are important, especially given that this GRACE-derived approach is now widespread in applied hydrology. This detailed analysis for such a large-scale integrative method is especially pivotal when the objectives are, as proposed in this study, to characterize finer processes poorly discussed in the literature (i.e. differential recharge rates depending on geological heterogeneities, and relative disconnection between the recharge front and the water table decline at the outskirt of intensive agricultural areas).
Reviewer #1 judges that the two features ("effects of intensive agriculture and geological heterogeneity") that are put forward in the title *"are not so deeply considered in the work"*. These two points are however original outcomes of this study that are also highlighted by Reviewer #2. As an example, we clearly show that intensive agriculture is by far the main output of the water budget of the Saq aquifer and that the associated irrigation return flow ends up to be an input as significant as the natural recharge. Such process was not accounted for by the previous GRACE-derived water budget studies (see for example Fallatah et al 2019).
So, we think that the title should let it show these original features specific to the studied site. We agree that the tool/method itself could appear verbatim in the title along with the main results and processes studied. We propose to modify the title (line 1) such a way to take

account of these two aspects as follow: "Influences of intensive agriculture and geological heterogeneity on the recharge estimate of an arid aquifer system (Saq-Ram, Arabian Peninsula) inferred from GRACE data". Maybe the editorial board would have advices regarding this matter?

**2 Specific comments**

*1. Lines 87 to 97. The resolution which can be reached with the remote sensing approach should be discussed. Also the extension of the basin which can be properly studied with satellite data should be discussed. For instance, at line 96, it would be useful to explicitly state how large are the mentioned regional aquifers.*

"Without the use of specific downscaling approaches, these selected aquifers are usually larger than $0.1 \times 10^6$ km² given the 3°×3° native spatial resolution of the GRACE data (filtered up to 1°×1° for some products; Landerer and Swenson, 2012; Wiese et al., 2016)." have been added lines 98, as well as the surfaces of the mentioned regional aquifers line 111: "from $0.6 \times 10^6$ km² to $2 \times 10^6$ km²".

The extension of each basin have been also added to the discussion section 4.2 (line 551): "Murzuq Basin in Libya ($0.6 \times 10^6$ km²; Mohamed and Gonçalvès, 2021 using GRACE data from Bonsor et al., 2018); Nubian Sandstones System (NSAS) covering Egypt, Libya, Tchad and Soudan ($2.1 \times 10^6$ km²; Mohamed et al., 2017); North Western Sahara Aquifer System (NWSAS) in Algeria, Tunisia and Libya ($1.2 \times 10^6$ km²; Mohamed and Gonçalvès, 2021); Tindouf basin in Algeria and Morocco ($0.3 \times 10^6$ km²; Gonçalvès et al., Submitted); and Djeffara basin in Tunisia and Libya ($0.1 \times 10^6$ km²; Gonçalvès et al., 2021)"

*2. Line 214. The averaged water demand for agricultural purposes is given with three or four significant digits. I do not think this is physically acceptable, also taking into account the estimates of uncertainties which are explicitly given later for similar quantities.*

The modifications have been addressed line 247:
"In addition, Alhassan et al. (2016), and Chowdhury and Al-Zahrani (2013) provided regional values of the agricultural water demand for the 2009-2012 period yielding an average $7.7 \times 10^9$ m³ yr⁻¹ for the Saq aquifer. Domestic withdrawals for 2003 are given by the BRGM and Abunayyan Trading Corp. (2008) for each region of Saudi Arabia yielding $300 \times 10^6$ m³ yr⁻¹ for the entire Saq aquifer in that year. The 2003 industrial water demand is also reported by the same authors at about $20 \times 10^6$ m³ yr⁻¹.
Finally, Jordan groundwater pumping data are given by region in 2015 and 2017 by the Jordan Water Sector Facts and Figures (Almomani et al., 2015; url: www.mwi.gov.jo) both reporting about $300 \times 10^6$ m³ yr⁻¹ for the Jordanian part of the Saq-Ram Aquifer System."

*3. Figure 3. These data deserves further comments. Any remarks about the increasing discrepancies among the products by JPL, CSR and GSFC for TWS, which is clearly apparent after 2012? Which regression is applied to TWS time series? The linear trends for SWS are not very appropriate, as these time series show, at least, a three-phase behavior: great variability from 2002 to 2006; small variations and values close to zero or poorly negative from 2007 to 2018; high variability and positive values after 2019.*

We agree that these data should be commented. The following paragraph will be added line 357:

"While the time series of the three GRACE solutions (Figure 3a) clearly show a significant decrease of the *TWS* chiefly due to an increasing groundwater deficit (no permanent surface water bodies over the studied domain), the GLDAS products reveal a three-phase behavior of the *SWS* signal (Figure 3b) which can be related to variations of the climatic inputs of the respective models: a great variability from 2002 to 2006; small variations and values close to zero, or poorly negative, from 2007 to 2018; and high variability and positive values after 2019. The increasing discrepancies among the GRACE-JPL, -CSR and -GSFC products for *TWS*, which is clearly apparent after 2012, are mainly due to the diverse shape and size of the Mascons, and the various methods of eliminating signal leakage effects used by the three respective computing centers. Lenczuk et al. (2020) acknowledge that the GRACE-GSFC product differs the most from the others."

Regarding the regressions presented Figure 3, these were just to better illustrate the trends but, in fact, the GWS signal (i.e. TWS minus SWS) is computed on a monthly basis, and then a polynomial fit is applied on the resulting GWS time-series in order to assess the long-term trend. Thus, regressions which introduce unnecessary confusion have been removed from the Figure 3. The polynomial fit applied to the resulting GWS signal was mentioned in the following section (2.4 Methods), but have been more clearly commented line 383: "Upon identification of a long-term trend (18 years in our case) of the *GWS* signal (using a 4$^{th}$ order polynomial regression to filter out the seasonal signal), *ΔGWS* is calculated and can be introduced in an overall groundwater balance equation to identify one unknown (e.g. recharge) if all other fluxes are known.".

*4. Sections 4.3 & 4.4. I appreciate this discussion. However, the two sections could be merged in a better way. At the beginning of section 4.3 it is stated that the transit time of water through the vadose zone is about 350 years; taking into account porosity and saturation, the recharge times should be even longer. This immediately rises the concern about the fact that the dynamics of the unsaturated zone and of the aquifer recharge seems to be much slower than the variability of the gravity field. Furthermore, the time series considered here are 20 years long, which is a time interval much shorter than 350 years. At line 541, it is stated that 12 years of GRACE data are required to obtain the long-term average recharge. This seems to be in contrast with the previous comments, doesn't it?*

Indeed, the transit time of the infiltration through the vadose zone compared to the time period considered for the gravity data is a real concern. That is why we proposed these two discussions with a local application (section 4.3) and a general conceptual representation of the 'standard' GRACE application to hydrogeology (section 4.4) which cannot be merged since the considered scales and resulting interpretations are very contrasted.

As stated section 4.3: "the transit time of water across the vadose zone in this area (i.e. 70 m) is about 350 years". This is valid only for this sand dune area studied by Al-Sagaby and Moallim (2001) submitted to natural recharge only. However, this transit time would be much lower in agricultural areas since the water content and the total recharge (artificial + natural) would be much greater. Since we do not know what is the domain averaged water content of the vadose zone for the whole Saq-Ram aquifer, this transit time cannot be obtained at this large scale. Hence, the 350 years of transit time cannot be directly compared to the time series of gravity data (20 years).

On the other hand, section 4.4 answers your concerns by demonstrating that with large transit times of the water in the vadose zone (presumably in between 0 and 350 years) it is difficult

to attribute an annual GRACE-derived recharge to a specific year since the GRACE-GLDAS approach on annual time-scale does not yield annual recharges, but annual infiltrations at the bottom of the GLDAS models (i.e. 2m depth). That is why we advise to use only a long-term trend of the Gravity signal (15 years) to obtain the infiltration rate characteristic of a steady-state (> 10 years) in the case of arid aquifers with thick vadose zone and mostly diffuse recharge process. This steady-state infiltration rate can be considered as a long-term recharge rate since it buffers the annual variations of the recharge. Indeed, even if the rain water takes hundreds of years to reach the water table surface, there is still older water reaching the it during the GRACE time frame (i.e. a sort of piston effect). One can assume that if the resulting natural recharge is valid now, it was more or less the same 350 years ago.

**3 Technical comments**

*1. I recommend to follow as carefully as possible the guidelines by BIPM about the use of SI units and the format with which values of physical quantities are written. In particular, I recommend to use parentheses to avoid confusion. I list here some examples.*
*(a) A space should be left between the value and the percentage symbol, e.g., 1 %.*
Done (lines 85, 123).

*(b) Line 81. "3-4%" should be substituted, possibly with "approximately 3.5 %". Similarly, at line 107.*
Done (lines 88, 122).

*(c) Line 109. "1.8 ± 0.3%" should be substituted with "(1.8 ± 0.3) %". Similarly, at lines 115, 345, 388, 418, 419, 445, 454, 459, 562.*
Done (lines 30, 124, 130, 451, 506, 552, 553, 584, 598, 603, 704)

*(d) Lines 187 & 188. "30 and 90 × 106m3yr−1" should be substituted, possibly with "30 × 106 m3 · yr−1 and 90m3 · yr−1". Similarly at line 220.*
"ranging between $30 \times 10^6$ m$^3$ yr$^{-1}$ and $90 \times 10^6$ m$^3$ yr$^{-1}$" (line 214).
"both reporting about $300 \times 10^6$ m$^3$ yr$^{-1}$ for the Jordanian part of the Saq-Ram Aquifer System" (line 254)

*(e) Line 312. "210 ± 30 × 106m3yr−1" should be substituted with "(210 ± 30) × 106 m3 · yr−1". Similarly at lines 325, 330, 355, 387, 462, 463, 465, 488, 489, 541, 559 to 561.*
Done for (xxx ± xxx) × $10^6$ m$^3$ yr$^{-1}$ and for (xx ± xx) mm yr$^{-1}$ (lines 34, 417, 418, 431, 458, 459, 466, 606, 607, 608, 609, 630, 631, 699, 701, 702, 703).

*2. Line 51. Although Bierkens and Wada (2019) is a review paper, it could be fair to add citations of some of the seminal papers on this subject.*
We agree. The citation of the first global assessment of groundwater depletion (Wada et al 2010) have been added (line 57): "When aquifer recharge is lower than withdrawals, this depletion constitutes permanent groundwater mining (Bierkens and Wada, 2019; Wada et al., 2010)."
Wada, Y., van Beek, L. P. H., van Kempen, C. M., Reckman, J. W. T. M., Vasak, S., and Bierkens, M. F. P.: Global depletion of groundwater resources, Geophys. Res. Lett., 37, https://doi.org/10.1029/2010GL044571, 2010.

*3. Line 54. Since the resolution of GRACE data and of the products used in this work are given in degrees, it would be useful to include the extension of the basin in latitude and longitude. This can be done here or in section 2.1.*

"spanning 10° longitude and 8° latitude" have been added at the beginning of section 2.1 (line 152).

*4. Lines 151 & 152. A verb is missing in the sentence "The Climatic Research Unit... (over the period 1901-2019)", isn't it?*

The verb "is" was not missing, but the paragraph has been modified for clarity purpose: "Mean annual temperature is about 26°C ranging between 27 and 43°C in summer, and between 8 and 20°C in winter. Typical of other arid domains, recharge (when it occurs) is associated with intermittent rainfall events (de Vries and Simmers, 2002). Extracted from the Climatic Research Unit database (CRU; Harris et al., 2020), the long term annual average rainfall (AAR) over the studied domain is 55 mm yr$^{-1}$ (over the period 1901-2019), with a maximum of 80 mm in 1982, and a minimum of about 40 mm in 1978 and 2009 (Figure 2)."

*5. Lines 212, 318; legend of figure 2. "et al." should be substituted with "and McCluskey".*

These citations within the manuscript are correct, however an error was present in the references (due to my citation software). The reference has been corrected to (line 750): "Alhassan, A. A., McCluskey, A., Alfaris, A., and Strzepek, K.: Scenario Based Regional Water Supply and Demand Model: Saudi Arabia as a Case Study, Int. J. Environ. Sci. Dev., 7, 46–51, https://doi.org/10.7763/IJESD.2016.V7.739, 2016."

*6. Figure 2. Why only one time series is shown with error bars? How is the uncertainty estimated?*

This is explained in the dedicated section (3.1 Groundwater pumping; line 420):
"Using previously published data (see Sect. 2.2), we can also estimate a continuous agricultural pumping time series for the Saudi part of Saq-Ram Aquifer System (Figure 2). First, we computed linear regressions between the Othman et al. (2018) time series (grey curve in Figure 2) and (i) BRGM and Abunayyan Trading Corp. (2008) data (green curve), (ii) Alhassan et al. (2016), and Chowdhury and Al-Zahrani (2013) data (red curve) given per region of Saudi Arabia. Subsequently, as shown in Figure 2, the mean agricultural withdrawals and their uncertainties were obtained (black curve) by averaging these two regressions."

The data previously published were not given with associated uncertainties since it originally comes from the Ministry of Agriculture. So, uncertainties are only due to averaging the two regressions. References to Figure 2 have been added to this paragraph for clarity purpose.

*7. Lines 231 & 232. Expression "its 2004.0-2009.9 average value" should be substituted, possibly with "its average value from January 2004 to September 2009"*

That is the way the NASA-JPL literature defines gravity anomalies, but it is indeed confusing. It has been substituted by "gravity anomalies (i.e. gravity value of a given month minus the average value from January 2004 to December 2009)" (line 267), and by "The *SWS* anomalies were computed the same way as for the GRACE *TWS* anomalies, i.e. by subtracting the January 2004 to December 2009 average from each monthly value of the time series." (line 353).

*8. Figure 3. For both plots, I could not find an explicit definition of the reference equivalent water thickness, which corresponds to a null value.*

GRACE does not provide absolute gravity measurements, only measurements relative to an average value. Line 267 have been modified into: "gravity anomalies (i.e. gravity value of a given month minus the average value from January 2004 to December 2009)". And "Monthly gravity anomalies, denoted terrestrial water storage (*TWS*), are expressed in water height (i.e. Equivalent water thickness)" have been added line 270. We computed GLDAS SWS the same way to be consistent (as stated line 353-355). Hence, a zero 'Equivalent water thickness' means that the gravity value of a given month equals the average from January 2004 to December 2009. The term "anomalies" was also added to the caption of Figure 3 as well as "Monthly time series" (line 305).

*9. Lines 264 & 265. "(GLDAS data are available at https://ldas.gsfc.nasa.gov/gldas)" should be moved before, possibly at the end of the first sentence of this paragraph.*
"GRACE/GRACE-FO Mascon data are available at http://grace.jpl.nasa.gov" have been moved line 289. And "GLDAS; data available at https://ldas.gsfc.nasa.gov/gldas" have been moved at the end of the first sentence of the paragraph (line 345).

*10. Line 290. Which polynomial fit is used? I mean, second, third, or higher degree? I do not see ΔGWS in Figure 3.*
"(using a 4th order polynomial regression to filter out the seasonal signal)" has been added line 383. Adding the GWS signal to Figure 3 would overlap the TWS signal since SWS variations are very low compared to TWS. It would therefore require another graph which we don't think is essential.

*11. Line 343. In which sense "needed"?*
This is a common practice in agriculture: an excess of the dose of irrigation required by crops is necessary to avoid the concentration of pollutants (most of the time salts) which reduces soil fertility.

**RC2 - Anonymous Referee #2 – May 10, 2022**

**Review of the manuscript egusphere-2022-22**

*This is a review for the article: "Influence of intensive agriculture and geological heterogeneity on the recharge of an arid aquifer system (Saq-Ram, Arabian Peninsula)" by Seraphin and colleagues. Seraphin and colleagues present a method that combines GRACE satellite products with Global land data assimilation model outputs into a simple regional water balance model for the estimation of regional groundwater recharge rates. Furthermore, the importance of artificial recharge from irrigation return flows is evaluated and compared to the estimated natural groundwater recharge rate, as is the importance of recharge over the comparably limited geographic extents of volcanic deposits (with both artificial recharge and recharge over volcanic deposits being hugely important for the aquifer system). The study is very well researched, presented and written, and can provide guidance to similar estimations for other data scare regions. I have only few minor concerns that I list below. Once these points have been addressed, I recommend moving forward and accepting the article for full publication in HESS.*

**Minor Comments:**

**General:**

*- The abstract is generally well written, clearly describes the goals, the data sources as well as the results. However, no details on the applied method are provided. I suggest reducing mentioning the different datasets and GRACE solutions in such detail, as writing out the different names plus providing the abbreviations consumes way too much space. The freed up space I suggest using for a sentence on the applied methodology, for example, linking to the first sentence of section 2.4: "The data was used to build a regional-scale water mass-balance and estimate recharge from variations in groundwater storage"*

Since the different versions of the products are listed in Table 1, we shortened lines 291-294 and 349-352 by removing the version numbers and writing the products names and citations in line. However, even if this may seem fastidious, the mention of the different datasets and their associated versions and citations are important to us since both GRACE and GLDAS are often updated with new measurements, but also with new treatments that can cause differences between results obtained for the same study area by using different versions of the datasets. In fact, this is one of the differences (among others) that may explain discrepancies with the study by Fallatah et al. (2019) mentioned in the introduction.

The suggested sentence "The data is used to build a regional-scale water mass-balance and estimate recharge from variations in groundwater storage (*ΔGWS*)." has been added at the beginning of section 2.4.

*- I appreciate that uncertainties are provided for every number, a sign of a thorough analysis and something that is too often absent missing in similar analyses. Nevertheless, the uncertainties seem very small for satellite based assessments of recharge over such a large domain. Based on the numbers provided (i.e., 4.4 +/- 2.6%), Seraphin et al suggested that recharge rate of the entire Saq-Ram Aquifer lies between ~2-7% of the average annual rainfall. Isn't this range a little narrow considering that the datasources are satellite based data and global land data assimilation model outputs? In the cited study by MacDonald et al*

*2021, who synthesised recharge rates in arid africa based on more local and therefore generally more accurate methods for the studied regions, assume recharge rates of 3.3 +/- 5.5% of the annual average rainfall and thereby provide a more conservative range of ~0-10%. What I want to say is that I believe that the uncertainty estimates are too small and neglect some intrinsic uncertainty in the source products used for these calculations. This being said, I believe that the order of magnitude of the estimated recharge is very reasonable. To conclude, I expect that the authors add a thorough discussion of how reasonable these uncertainty estimates are given the uncertainty in the source products and the applied method.*

We agree that the uncertainty that we obtain for the natural recharge rate can seem narrow considering a satellite-based approach. However, this approach is integrative and probably best suited to the large studied domain. By contrast, the synthesis reported by MacDonald et al. (2021) gathers various local methods known to be less integrative (mostly chloride mass balance and environmental tracers). As explained lines 126-130, we averaged the results from multiple areas in Africa (with sometimes several studies for a same site) including seven of them yielding a zero recharge (out of 23 with rainfall < 150 mm yr$^{-1}$). Thus, the resulting average uncertainty is greater (i.e. 3.3 +/- 5.5%), and cannot be compared with the one from a single and more or less homogeneous study site using a different approach (i.e. 4.4 +/- 2.6% using the gravity-based water budget).

Moreover, GRACE products do not provide associated uncertainties for the raw data used here (they do so for the JPL's scaling factors for example, but we did not use those since it would be incoherent with the study of such an arid domain with TWS variations mainly driven by groundwater mass variations as stated lines 295-300). And even if they were providing uncertainties, it would still be lower than averaging the products as we did (i.e. JPL and CSR).

But, to illustrate this, we propose to add a comment about uncertainties at the beginning of the discussion section (line 510), and more specifically about Blazquez et al. (2018) exploring the uncertainty in GRACE estimates:

"This study provides an estimate of the 2002-2019 domain-average natural recharge with associated uncertainty (one sigma) accounting for temporal variations in natural discharge, groundwater pumping and irrigation return flow. The uncertainties associated with the calculation of the *ΔGWS* long-term trends with the GRACE and GLDAS products have also been considered. Errors associated with GRACE measurements could not be accounted for as they are not provided with the raw Mascons data (i.e. before the application of unwanted scaling factors). However, Blazquez et al. (2018) investigated the uncertainty of raw GRACE data by solving a global water budget using trends in ocean mass, ice loss from Antarctica, Greenland, arctic islands and trends in water storage over land and glaciers. The authors estimated a 0.27 mm yr$^{-1}$ uncertainty for the GRACE data, a figure significantly lower than the uncertainties of the *ΔGWS* trends used in this study (Table 1)."

A Blazquez, B Meyssignac, JM Lemoine, E Berthier, A Ribes, A Cazenave, Exploring the uncertainty in GRACE estimates of the mass redistributions at the Earth surface: implications for the global water and sea level budgets, Geophysical Journal International, Volume 215, Issue 1, October 2018, Pages 415–430, https://doi.org/10.1093/gji/ggy293

*- Lines 191-193: Related to the above comment, rather than just assuming that unquantified outlets constitute minor outflows and then neglecting them, I suggest considering the impact of such outflows along the boundaries of the system quantiatively by adding a term to the*

*water balance and extending the uncertainty analysis. This can be done very quickly and would put a number on that assumption, rather than neglecting such terms. The beauty of a simple water balance analysis is that such terms can easily be considered qualitatively. This could provide yet more realistic uncertainty estimates and help in resolving the aforementioned issue.*

We believe that adding an unknown random natural discharge to the water budget would artificially raise the uncertainty on the recharge and would be less rigorous than ignoring it. Indeed, this outflow (towards the southeastern Khuff aquifer) could easily be null (or even negative, i.e. inflow). This small limit is indeed located next to the Al Qasim area which is known to be the one presenting the most intensive groundwater withdrawals. This creates a large drawdown cone almost reaching the eastern aquifer limit. Moreover, the total natural discharge fluxes being two orders of magnitude lower than groundwater pumping (Table 1), accounting for this hypothetical outflow would have an insignificant impact on the resulting estimate of natural recharge.

However, to answer this legitimate question, we suggest to add the following sentence at the end of section 2.2.2 (line 222):
"Finally, with regard to historical piezometric maps of the Saq aquifer (Sharaf and Hussein, 1996; Lloyd and Pim, 1990), it can be assumed that the southeastern limit with the Khuff aquifer is likely inactive given the large drawdown cone created by the intensive pumping of the Al Qasim area."

- Sharaf, M. A. and Hussein, M. T.: Groundwater quality in the Saq aquifer, Saudi Arabia, Hydrol. Sci. J., 41, 683–696, https://doi.org/10.1080/02626669609491539, 1996.
- Lloyd, J. W. and Pim, R. H.: The hydrogeology and groundwater resources development of the Cambro-Ordovician sandstone aquifer in Saudi Arabia and Jordan, J. Hydrol., 121, 1–20, https://doi.org/10.1016/0022-1694(90)90221-I, 1990.

*- I miss a discussion of the importance of ephemeral and intermittent streams in arid regions. These are often the main sources of recharge in arid regions as they collect and distribute the rainfall rapidly throughout the system, making infiltration available also to regions where it didn't rain locally. As stated in the beginning of section 2.4, permanent surface water bodies are almost completely absent from the Saq-Ram aquifer system, but surely intermittent systems are not. In other words, what happens to all the rainfall that doesn't form recharge prior to it being evaporated or transpired (i.e., ~95% of the AAR, according to the calculations in this study)? Before that water evaporates or is consumed and transpired by vegetation, it certainly forms intermittent stream networks. I suggest adding a short paragraph on their importance for (eco)hydrology and especially groundwater recharge in arid regions, supported by the at least the three references listed below, and drawing a link to intermittent streams on the arabian peninsula and the Saq-Ram Aquifer system. This would nicely round off the discussion of the importance of artificial recharge from agriculture and of the geology on the regional recharge. Suggested references:*
*Bourke et al., 2020, doi: 10.1002/wat2.1504*
*Dogramaci et al., 2015 , doi: 10.1016/j.jhydrol.2014.12.017*
*Schilling et al., 2021, doi: 10.1029/2020WR028429*

Thank you for the useful references. We can add this comment at the beginning of the discussion section (line 521):

"Even if the Saq-Ram domain is devoid any permanent surface water bodies, ephemeral streams are known to be important for (eco)hydrology and local groundwater recharge in arid regions (Shanafield et al., 2021; Dogramaci et al., 2015; Schilling et al., 2021). However, runoff coefficients were estimated at about 1% in the region (Al-Hasan and Mattar, 2013) while more than 90% of this runoff is lost by evaporation in the low lands. Thus, accounting for recharge redistribution through ephemeral streams in the domain-average water budget of the large Saq-Ram aquifer system would be quantitatively insignificant."

Al-Hasan, Abdul Aziz Saleh and Yousry El-Sayed Mattar. "Mean runoff coefficient estimation for ungauged streams in the Kingdom of Saudi Arabia." Arabian Journal of Geosciences 7 (2013): 2019-2029.

*- Lines 270-273: This is not well explained and it's completely unclear to me which polygons were used to spatially average the GRACE and GLDAS products over the studied domain. In Figure 3, which is referenced here, no polygons or maps are provided. Explain.*

How anomalies were computed (both with GRACE and GLDAS datasets) was also unclear for Reviewer #1 so the first sentence of this paragraph (line 353) can be edited as:
"The *SWS* anomalies were computed the same way as for the GRACE *TWS* anomalies, i.e. by subtracting the January 2004 to December 2009 average from each monthly value of the time series."
Regarding the spatial average, we weighted the monthly spatial average of each signal by the proportion of each GRACE/GLDAS polygon within the domain (cut through by the aquifer system limit). This is stated in the next sentence (line 355):
"Both the GRACE products and GLDAS solutions were spatially averaged using the surface weight of each polygon within the Saq-Ram Aquifer domain (Figure 3)."
So, a polygon fully within the domain has a weight of 1, while a polygon 30% within the domain has a weight of 0.3 in the computation of the spatial domain averages.
We tried to provide the GLDAS/GRACE polygons on figure 3, but the fact that each product has a different size and shape of grid cell (listed section 2.3) makes the later unreadable, and adding multiple maps for this purpose seemed unnecessary (especially since this is an easily accessible information: https://grace.jpl.nasa.gov/data-analysis-tool, and https://ccar.colorado.edu/grace/gsfc.html).

**Figures:**

*- Figure 1: provide coordinate reference system*

The caption of the Figure 1 (line 165) can be modified as such:
"Context map of the Saq-Ram Aquifer System (WGS84 coordinates shown by straight dotted lines every 5 degree; Shorelines and country borders extracted from Wessel and Smith, 1996; Administrative regions extracted from www.gadm.org)"

*- Figure 2: from the legend and the caption it is not clear what the different lines refer to. Extend the caption to provide more information than just references without any additional info. It should be clear from the caption alone what is meant, the reader should not have to go and dig through the manuscript for the explanation of those references first.*

The colored lines correspond to different sources of agricultural withdrawal data. The figure caption (line 257) can be modified as:

"Annual average rainfall (Climatic Research Unit; mm yr$^{-1}$) of the Saq-Ram Aquifer System and agricultural withdrawal (from different sources represented by colored lines; $10^6$ m$^3$ yr$^{-1}$) of its Saudi part (except for Othman et al.'s (2018) data corresponding to Al-Qassim, Ha'il and Al-Jouf regions of Saudi Arabia)."

*- Figure 3: Change axis titles to all lower case titles or use the already introduced abbreviations right away.*

Done.
And caption of the Figure 3 (line 305) can be modified for clarity purpose:
"Monthly times series of (a) the GRACE-JPL, -CSR, -GSFC terrestrial water storage anomalies (TWS; mm), and (b) the GLDAS-VIC, -CLSM, -NOAH soil water storage anomalies (SWS; mm) of the Saq-Ram domain."

*- Figure 7: A very nice analysis and presentation*

Thank you.

**Abbreviations:**

*- The Kingdom of Saudi Arabia (KSA) is sometimes called 'Saudi Arabia', sometimes with the abbreviation KSA or sometimes simply referred to as 'Saudi'. Double check and make the naming consistent throughout the manuscript.*

We will only use the terms 'Saudi Arabia' and 'Saudi' (referring to what "belongs" to Saudi Arabia) instead of 'KSA' (lines 179, 190, 191, 194, 195, 234, 250, 260, 425, 434, 448, 569).

*- The GRACE abbreviation is introduced at least three times: once in the abstract, once in the intro, once in the methods. Introduce it once in the intro, that is sufficient. Remove the introduction of the abbreviation in the abstract to save space for more relevant info.*

Done.

*- Other abbreviations such as terrestrial water storage (TWS), GWS and SWS are also introduced multiple times (three or four times at leas). Moreover, the full names are written with capital first letters in the figures' axis titles (e.g, Figure 3), rather than without capital letters or with the abbreviations. Avoid introducing abbreviations so many times and make the naming consistent throughout the manuscript.*

Done.

But, if it is possible, we would like to keep the definition of TWS and SWS both in the data section (2.3) and the method section (2.4) so the reader have a clear understanding of the methodology without having to refer to the prior data section (2.3).

**References:**

*- The intro is a little light on recent references, particularly on available methods for groundwater recharge quantification (in arid regions). Since all methods are subject to different sources of considerable uncertainty, it would be good to provide more references and to direct the reader to this information. I would suggest adding the following references to lines 62-65:*
*Shanafield and Cook, 2014, doi: 10.1016/j.jhydrol.2014.01.068*
*Banks et al., 2020, doi: 10.1016/j.jhydrol.2020.125753*

Done. Added line 70.
Thank you for providing such useful references.

*- For the discussion of the importance (and dominance) of intermittent streams on groundwater recharge in arid regions, see comment above*

Acknowledged.

*- Lines 249-252: provide a reference for this statement*

Done. One statement (line 296) can also be added to justify that the use of scale factors is not suitable for this study:
"In fact, these downscaling factors are based on the mass distribution calculated by land surface model (LSM) accounting for surface and subsurface water transfers (Landerer and Swenson, 2012) while *TWS* variations in such arid regions are expected to be chiefly controlled by groundwater mass variations. Moreover, as stated by the authors, the use of such gain factors is not suitable to quantify trends."

Landerer, F. W. and Swenson, S. C.: Accuracy of scaled GRACE terrestrial water storage estimates, Water Resour. Res., 48, https://doi.org/10.1029/2011WR011453, 2012.

**RC3 - Anonymous Referee #3 – May 18, 2022**

**Review Report: egusphere-2022-22**

*The paper "Influence of intensive agriculture and geological heterogeneity on the recharge of an arid aquifer system (Saq-Ram, Arabian Peninsula)" presents a water budget estimation of regional groundwater recharge flux for an arid aquifer system.*
*The approach is well driven and the paper is really well written. I do have three remarks.*

*1) The authors claim that there is a strong spatial heterogeneity of groundwater recharge over the domain. However, such heterogeneity is barely discussed and not illustrated. A representation of the variation of the estimated recharge flux over the studied region would be a great asset. It would be interesting to have such a map and to discuss about the heterogeneity of the estimated recharge in the results. Moreover, it is not clear for now how the water budget has been computed and how the other component of the water budget has been assimilated and combined with the satellite-based products. Are the authors computed the water budget on a regular grid over the whole region? The method section (2.4) is more a theory section and is missing information about how the water budget was practically computed.*

Indeed, we have not been able to provide a mapping of the recharge. This is because the GRACE-derived approach is highly integrative, in addition to the coarse resolution of the raw data (i.e. 3° x 3°, giving only 10 meshes partially in the domain with zero totally included). So, such mapping could not be relevant with this approach, and would be even very complicated with other approaches since most of the required data is not spatialized if it exists. Thus, we stated explicitly that we investigate the regional-scale water mass-balance (e.g. line 371 in the method section) using only domain averages of each contribution. However, heterogeneities are addressed through the discussion in section 4.2 (i.e. a natural recharge fifteen times more effective on fractured basaltic deposits than on sedimentary formations) and in section 4.3 (i.e. a recharge temporally null at the outskirt of the large irrigated areas due to a recharge front velocity much lower than the thickening of the unsaturated zone resulting from the nearby pumping).

*2) The authors present an estimation of the regional groundwater recharge about 2.4 mm.y-1 (with an uncertainty of 1.4 mm.y-1). Beside that, it is unclear if all the uncertainties presented in the paper are for 1 or 2 σ. Knowing that each component of the water budget contained large errors, it is not clear to me how the authors can have such a precise estimate of groundwater recharge. I would rather think that estimation of such a small recharge flux would be very challenging as the cumulative effects of the errors in each water budget components are also large. It would be great to have a discussion about what is really quantified in the "uncertainties" and what are the major limits of such estimation of recharge flux with a large-scale water balance approach. Also, what are the effects of boundary conditions (lines 187, 188) in the calculation of natural discharge?*

Also required by Rewiever #2, we propose to add some information about uncertainties at the beginning of the discussion (section 4, line 510):
"This study provides an estimate of the 2002-2019 domain-average natural recharge with associated uncertainty (one sigma) accounting for temporal variations in natural discharge,

groundwater pumping and irrigation return flow. The uncertainties associated with the calculation of the $\Delta GWS$ long-term trends with the GRACE and GLDAS products have also been considered. Errors associated with GRACE measurements could not be accounted for as they are not provided with the raw Mascons data (i.e. before the application of unwanted scaling factors). However, Blazquez et al. (2018) investigated the uncertainty of raw GRACE data by solving a global water budget using trends in ocean mass, ice loss from Antarctica, Greenland, arctic islands and trends in water storage over land and glaciers. The authors estimated a 0.27 mm yr$^{-1}$ uncertainty for the GRACE data, a figure significantly lower than the uncertainties of the $\Delta GWS$ trends used in this study (Table 1)."

- A Blazquez, B Meyssignac, JM Lemoine, E Berthier, A Ribes, A Cazenave, Exploring the uncertainty in GRACE estimates of the mass redistributions at the Earth surface: implications for the global water and sea level budgets, Geophysical Journal International, Volume 215, Issue 1, October 2018, Pages 415–430, https://doi.org/10.1093/gji/ggy293

However, the probability associated to the uncertainties was indeed missing. Thus, we added "one sigma" to the beginning of the suggested paragraph above, as well as in the Table 1 and its caption (lines 495-499).

The effects of boundary conditions have been also addressed by Reviewer #2 so we invite you to refer to our answers to the second and third General comments of Reviewer #2 which, we hope, will respond to these legitimate questions. Thereby, we suggested to add this sentence at the end of section 2.2.2 (Line 222):
"Finally, with regard to historical piezometric maps of the Saq aquifer (Sharaf and Hussein, 1996; Lloyd and Pim, 1990), it can be assumed that the southeastern limit with the Khuff aquifer is likely inactive given the large drawdown cone created by the intensive pumping of the Al Qasim area."
- Sharaf, M. A. and Hussein, M. T.: Groundwater quality in the Saq aquifer, Saudi Arabia, Hydrol. Sci. J., 41, 683–696, https://doi.org/10.1080/02626669609491539, 1996.
- Lloyd, J. W. and Pim, R. H.: The hydrogeology and groundwater resources development of the Cambro-Ordovician sandstone aquifer in Saudi Arabia and Jordan, J. Hydrol., 121, 1–20, https://doi.org/10.1016/0022-1694(90)90221-I, 1990.

*3) In section 4.4, the authors are discussing the focused groundwater recharge. I do miss a complement discussion about the effects/importance of the ephemeral stream (so-called wadi systems) in groundwater recharge processes at this scale. Are these systems quantified in the approach or ignored. In both cases, it is not clear.*

This was also addressed by Reviewer #2 in its third general comment. We propose to add the following paragraph at the beginning of the discussion section (line 521):
"Even if the Saq-Ram domain is devoid of any permanent surface water bodies, ephemeral streams are known to be important for (eco)hydrology and local groundwater recharge in arid regions (Shanafield et al., 2021; Dogramaci et al., 2015; Schilling et al., 2021). However, runoff coefficients were estimated at about 1% in the region (Al-Hasan and Mattar, 2013) while more than 90% of this runoff is lost by evaporation in the low lands. Thus, accounting for recharge redistribution through ephemeral streams in the domain-average water budget of the large Saq-Ram aquifer system would be quantitatively insignificant."

- Al-Hasan, Abdul Aziz Saleh and Yousry El-Sayed Mattar. "Mean runoff coefficient estimation for ungauged streams in the Kingdom of Saudi Arabia." Arabian Journal of Geosciences 7 (2013): 2019-2029.

**Minor comments**

*# In the abstract, I think the JPL, CSR, GSFC, VIC, etc. might be removed to facilitate the comprehension.*

We modified this part of the abstract as follows:
"The three existing GRACE solutions were tested for their local compatibility to compute groundwater storage variations in combination with the three soil moisture datasets available from the Global Land Data Assimilation System (GLDAS) land surface models."

*# The authors use the word "global" (e.g. line 31, 75 etc.) several times in the text. Is the word "large-scale" or "regional" would better fit? This study is not at a global scale.*

This use of "global" can be indeed misleading. We modified this by:
"Beyond the regional-scale approach proposed here" line 31.
"considering the regional water table decline initiated in the mid-1980s" line 82.
"regional-scale mass-balance equations" line 110.
"Leading to domain-averaged values for the groundwater fluxes, the integrative approach proposed here" line 139.

*To sum-up, the present paper is an interesting study lacking of some clarity in the approach and discussion about the reliability of the proposed estimates of groundwater recharge. The scientific quality is good but the scientific significance is not as well because the authors are using a well-established method and I do not see the real contribution to scientific progress for such a study. Maybe the authors can be better explicit about the real significance of such a study. The figures are good quality overall.*
*This manuscript can be accepted with some minor revisions in my opinion.*

---

## Author Response (AR2)

**Report #2**

*Submitted on 25 Jul 2022*

Note that the **line numbers** are **based** on the **"revised_v2_marked.pdf"** version of the manuscript (line numbers changed with Microsoft Word's revision tracking mode).
- *Reviewer comments in italic font*
- Answer to reviewer comments in regular font
- Modifications of manuscript in red

**Anonymous referee #1**

*The manuscript revision has not considered the referees' comments in a fully satisfactory way.*

*I think that an appropriate and thorough discussion of scaling issues is still missing and this negatively impacts on the reliability of the remarks about the effects of intensive agriculture and geological heterogeneity on groundwater recharge. In my opinion, the concerns by Referee #3 ("...lacking of some clarity in the approach and discussion about the reliability of the proposed estimates of groundwater recharge... the authors are using a well-established method and I do not see the real contribution to scientific progress for such a study") have not been overcome.*

*Therefore, I am sorry, but I think that the manuscript cannot be considered for publication on a high-quality international scientific journal like HESS, unless it is heavily revised.*

*Below, I provide a couple of specific comments and a couple of technical comments.*

Acknowledging this severe appraisal of our study, it is however difficult to fully respond to the criticisms of Referee #1 as they flag general "scaling issues" without specifying which issues. Referee #1 cites Referee #3's initial review, whose comments are addressed in the manuscript revised version, namely:

- Better characterizing uncertainties
- Clearing the southeastern boundary condition inactivity (also illustrated by Figure 2 of this document)
- Discussing the effects and importance of ephemeral streams

The final conclusion of Referee #3 was: "*This manuscript can be accepted with some minor revisions in my opinion*", which means that his / her concerns were only secondary and that this study is expected to be published following minor revisions. Minor revisions are addressed in the second version of the manuscript and we believe that this should be taken into account.

Perhaps Referee #1's comment about "scaling issues" questions the low resolution of GRACE data used to constrain the water budget of a large aquifer system, and specifically its modern recharge. Although GRACE data only allow domain-average estimates, we believe that this approach constitutes a breakthrough in hydrology and remains a worthwhile estimate, as

evidenced by the numerous recent publications on this subject in "high-quality international scientific journal"(Scanlon et al., 2016, 2019, 2021; Sun et al., 2020; Fallatah et al., 2019; Bonsor et al., 2018; Mohamed et al., 2017; Fallatah et al., 2017; Rodell et al., 2018; Richey et al., 2015).

**SPECIFIC COMMENTS**

*1) Lines 49 to 51. The scientific literature on "groundwater sustainability" or "water budget" has been unacknowledged. For instance, Bredehoeft et al. (1982) provide a discussion and older references relevant for this work.*

We believe that the reference cited in this comment (Bredehoeft et al., 1982), dealing with the maximum pumping capacity of hypothetical small aquifers, is not really directly relevant to the topic of our manuscript, because the authors studied 2 very specific cases: (i) a small island in the middle of a freshwater lake with head boundary conditions allowing its replenishment by lake water beyond a given pumping rate; and (ii) a small homogeneous closed basin (3000 km²) recharged by two rivers with a single pumping location and a single natural discharge location.

In contrast, arid aquifers lack open water surfaces (i.e. lakes, rivers…) that could replenish groundwater and balance pumping beyond local recharge, making the comparison with the above cases inadequate. Thus, groundwater mining is a common feature that must be considered in the management of groundwater resources in arid systems.

Conclusions on such hypothetical, small-scale and simplified cases as those presented in Brefehoeft et al. (1982) are not transposable to the Saq-Ram aquifer, one of the largest aquifer systems in the world (500 000 km²) that includes multiple layers, several inlets and outlets, strong heterogeneities, and complex interactions (e.g. irrigation return flow).

Nevertheless, the reviewer's concern about scientific literature on groundwater sustainability has raised the issue of giving a relevant definition of "groundwater sustainability" in our manuscript. We have therefore included the definition of Gleeson et al. (2020), whose paper is focused precisely on this subject. We suggest modifying the first paragraph of the introduction (Lines 45-56) as follows:

"Freshwater resources in arid regions of the world face growing pressure. Limited reserves, sporadic rainfall, droughts, agricultural production, increasing population and living standards are contributing to environmental and economic pressures. As defined by Gleeson et al. (2020): "groundwater sustainability is maintaining long-term, dynamically stable storage and flows of high-quality groundwater using inclusive, equitable, and long-term governance and management". Groundwater resources in arid zones have been heavily exploited for the past 50 years or so, in order to meet growing demands, which has led to overexploitation and local long-term depletion in many cases (Al-Zyoud et al., 2015; Othman et al., 2018). When aquifer recharge is much lower than withdrawals, this depletion can constitute permanent groundwater mining (Bierkens and Wada, 2019; Wada et al., 2010). In arid and semi-arid regions, this is a frequent phenomenon, in particular where large aquifer replenishment mostly occurred under past climatic conditions (so called "fossil aquifers")."

*2) Section 4.3. I downloaded Al-Sagaby and Moallim (2001) from the web site
https://www.osti.gov/etdeweb/servlets/purl/20224661. Is this the right paper? In that paper, I
was not able to find data, which supports the remarks of section 4.3. Therefore, important
information is missing. First, in order to discuss issues related to spatial scales, it is
necessary to give data about the extension of the Al-Qasim sand dune and on the thickness of
the vadose zone (I could not find support in the scientific literature to the estimate of 70 m; is
this a fault of mine?). Furthermore, data about the distance of this sand dune from
agricultural plots is missing.*

This is indeed the right paper (Al-Sagaby and Moallim, 2001). The poor precision of the data
provided in the paper led to making inferences, especially regarding the location of the site.
The authors only mentioned that "The field sampling was carried out at the KACST field
station located at Qassim (320-km northwest of Riyadh)". Therefore, we deduced that it must
be located in the sand dune area East of Buraydah (see attached map, Figure 1).

[Figure]

**Figure 1**: Estimated location of the KACST field station "320-km northwest of Riyadh" (Al-
Sagaby and Moallim, 2001) shown by the red line. The black line represents the contour of
the Saq-Ram aquifer, and the green hashed areas corresponds to agricultural areas.

Beyond this geographical estimate, the key topic discussed in section 4.3 is the comparison
between orders of magnitude of natural recharge and the decline rate of the groundwater table.

As illustrated in the piezometric contour map below (Figure 2), the entire Al-Qasim region is influenced by one of the largest drawdown cones worldwide (about 500 km diameter at its maximum). Thus, the exact location of the site studied by Al-Sagaby and Moallim (2001) has little impact on the demonstration because the sand dune area is, by definition, not an agricultural zone. Hence, only the natural recharge estimated by the authors (i.e. 1.8 mm/yr) is effective on the soil column. Using the average water content of 0.01%, this leads to a 'natural recharge front velocity' of 0.18 m/yr, well below the minimum water decline of 0.7 m/yr reported by BRGM and Abunayyan Trading Corp (2008).

[Figure]

**Figure 2**: Piezometric contour map of the Al-Qasim region extracted from Sharaf and Hussein (1996).

In order to clarify this in the manuscript, we suggest modifying the first two paragraphs of section 4.3 (Lines 527-539) as follows:

"Using the Al-Sagaby and Moallim (2001) study, it is possible to estimate the recharge velocity through a sand dune located in the Al Qasim region (within the Saq-Ram domain). An average natural recharge of 1.8 mm yr[-1] obtained by chloride mass-balance together with a mean measured water content on the vadose zone of 0.01% yields a local pore velocity equivalent to a 'natural recharge front velocity' of about 0.2 m yr[-1].

It is interesting to compare this recharge velocity with the water table decline velocity. By definition, this sand dune area is located away from any agricultural plot (i.e. zero artificial recharge by irrigation return flow) but within one of the largest drawdown areas worldwide (about 500 km diameter; Sharaf and Hussein, 1996) caused by intensive pumping. Considering a conservative 30 m water table decline in 45 years (BRGM and Abunayyan Trading Corp., 2008), a minimum 0.7 m yr$^{-1}$ decline is computed on the outskirts of this piezometric depression. This is significantly faster than the local natural recharge velocity of 0.2 m yr$^{-1}$, suggesting that the unsaturated zone is thickening faster than the percolation flows into it."

**TECHNICAL COMMENTS**

*1) Figure 2. The remark that "the data previously published were not given with associated uncertainties since it originally comes from the Ministry of Agriculture" should be added somewhere, either in the text or in the figure caption.*

The sentence "Since most of this previously published data comes from governmental entities, no associated uncertainty is provided with it" was added to the caption for Figure 2 (Lines 239-240).

*2) Line 420 & Table 1. In table 1 "(1 sigma)" is superfluous after "(standard deviation)". Moreover, at line 420, it should be preferred to mention "standard deviation" as a quantification of uncertainty, rather than the informal expression "1 sigma".*

This was required by Anonymous Referee #2 and #3 ("*it is unclear if all the uncertainties presented in the paper are for 1 or 2 σ.*"). In our opinion, this addition (i.e. one sigma) was relevant since the reviewers were surprised by the low uncertainty associated with our recharge estimate. We leave it to the editor to decide whether or not it should be mentioned in Table 1 and associated caption.

However, the wording "standard deviation" is now used instead of "one sigma" (Line 425) since it comes directly after Table 1 in the manuscript.

**REFERENCES**

Al-Sagaby, A. and Moallim, A.: Isotopes based assessment of groundwater renewal and related anthropogenic effects in water scarce areas: Sand dunes study in Qasim area, Saudi Arabia, International Atomic Energy Agency (IAEA), 2001.

Bonsor, H., Shamsudduha, M., Marchant, B., MacDonald, A., and Taylor, R.: Seasonal and Decadal Groundwater Changes in African Sedimentary Aquifers Estimated Using GRACE Products and LSMs, Remote Sensing, 10, 904, https://doi.org/10.3390/rs10060904, 2018.

Fallatah, O. A., Ahmed, M., Save, H., and Akanda, A. S.: Quantifying temporal variations in water resources of a vulnerable middle eastern transboundary aquifer system, Hydrological Processes, 31, 4081–4091, https://doi.org/10.1002/hyp.11285, 2017.

Fallatah, O. A., Ahmed, M., Cardace, D., Boving, T., and Akanda, A. S.: Assessment of modern recharge to arid region aquifers using an integrated geophysical, geochemical, and remote sensing approach, Journal of Hydrology, 569, 600–611, https://doi.org/10.1016/j.jhydrol.2018.09.061, 2019.

Mohamed, A., Sultan, M., Ahmed, M., Yan, E., and Ahmed, E.: Aquifer recharge, depletion, and connectivity: Inferences from GRACE, land surface models, and geochemical and geophysical data, Geological Society of America Bulletin, 129, 534–546, https://doi.org/10.1130/B31460.1, 2017.

Richey, A. S., Thomas, B. F., Lo, M.-H., Reager, J. T., Famiglietti, J. S., Voss, K., Swenson, S., and Rodell, M.: Quantifying renewable groundwater stress with GRACE, Water Resources Research, 51, 5217–5238, https://doi.org/10.1002/2015WR017349, 2015.

Rodell, M., Famiglietti, J. S., Wiese, D. N., Reager, J. T., Beaudoing, H. K., Landerer, F. W., and Lo, M.-H.: Emerging trends in global freshwater availability, Nature, 557, 651–659, https://doi.org/10.1038/s41586-018-0123-1, 2018.

Scanlon, B. R., Zhang, Z., Save, H., Wiese, D. N., Landerer, F. W., Long, D., Longuevergne, L., and Chen, J.: Global evaluation of new GRACE mascon products for hydrologic applications, Water Resources Research, 52, 9412–9429, https://doi.org/10.1002/2016WR019494, 2016.

Scanlon, B. R., Zhang, Z., Rateb, A., Sun, A., Wiese, D., Save, H., Beaudoing, H., Lo, M. H., Müller-Schmied, H., Döll, P., Beek, R., Swenson, S., Lawrence, D., Croteau, M., and Reedy, R. C.: Tracking Seasonal Fluctuations in Land Water Storage Using Global Models and GRACE Satellites, Geophys. Res. Lett., 46, 5254–5264, https://doi.org/10.1029/2018GL081836, 2019.

Scanlon, B. R., Rateb, A., Pool, D. R., Sanford, W., Save, H., Sun, A., Long, D., and Fuchs, B.: Effects of climate and irrigation on GRACE-based estimates of water storage changes in major US aquifers, Environ. Res. Lett., 16, 094009, https://doi.org/10.1088/1748-9326/ac16ff, 2021.

Sharaf, M. A. and Hussein, M. T.: Groundwater quality in the Saq aquifer, Saudi Arabia, Hydrological Sciences Journal, 41, 683–696, https://doi.org/10.1080/02626669609491539, 1996.

Sun, Z., Long, D., Yang, W., Li, X., and Pan, Y.: Reconstruction of GRACE Data on Changes in Total Water Storage Over the Global Land Surface and 60 Basins, Water Resour. Res., 56, https://doi.org/10.1029/2019WR026250, 2020.

---

## Author Response (AR3)

**Answer to Report #1**
*Submitted on 16 Oct 2022*

Note that the **line numbers** are **based** on the **"revised_v3_marked.pdf"** version of the manuscript.
- *Reviewer's comments in italic font*
- Response to reviewers' comments in normal type
- Changes to the manuscript in red

Anonymous referee #1

*In my opinion, this is an excellent technical report. I confirm my first impression about this work, i.e., the comparison of three GRACE solutions (JPL, CSR, GSFC) is interesting, but the manuscript lacks some innovative content from the methodologicl point of view, in order to be considered for publication on HESS. The specific application, even if it is dedicated to a very important hydrogeological structure, is of minor interest for HESS' readers, whereas it would be more adequate for an hydrogeological journal. This is confirmed also by the abstract, which is almost fully focused on the specific application, i.e., the analysis of the recharge of the Saq-Ram aquifer system.*

*I am sorry, because I realize that I did not explain in a very explicit way my concern about scale issues. However, the authors basically caught the concept. GRACE data have a poor resolution with respect to the typical scale lengths characterizing the heterogeneity of aquifer recharge, which is discussed in a less accurate way. Also, the discussion in section 4.3 refers to a structure that is considered at a scale length which is not consistent with GRACE resolution.*

Admittedly the GRACE approach presented in this study is regional whereas this section (4.3) involves a local recharge value as clearly indicated by its title. Praised by one of the reviewers (Anonymous Referee #1 - May 8, 2022 - Specific comment #4), this section highlights a feature poorly discussed in the literature (only one article to our knowledge): a relative disconnection of the infiltration front with the declining water table induced by intensive pumping. As an opening discussion, we used local data to calculate conservative recharge and piezometric decline velocities showing a temporarily zero recharge. Hence, we believe that this rarely highlighted feature provides an additional source of recharge heterogeneity likely to occur elsewhere (i.e. multiple similar drawdown cones induced by pumping and potentially exhibiting the same behavior at their periphery).

*Moreover, the clarifications given in the authors' response shows that the remarks of section 4.3 are based on literature information characterized by high uncertainty.*

While this article is fairly uncertain is terms of exact location of the studied site, it has been illustrated by multiple maps that the exact location is of little importance for the demonstration

since the whole sand dune area studied by Al-Sagaby and Moallim (2001) is under the influence of the 500km-diameter drawdown cone of Al-Qasim region (see our previous response Report #2 - July 25, 2022 - Specific comment #2).

*Therefore, I am sorry, but my overall opinion is that the manuscript cannot be considered for publication on HESS, mostly because I think that the innovative content is not sufficient for the ambitions of the journal.*

*However, if the editor's decision were different, I suggest to take section 4.3 off the manuscript and to fix the following technical details.*

In order to ease the publication process and since this opening discussion is not pivotal for the understanding of the study, we have removed it from the manuscript as recommended by the reviewer and the editor:

(Line 540) 4.3 Local recharge velocity and water table decline

Using the Al-Sagaby and Moallim (2001) study, it is possible to estimate the recharge velocity through a sand dune located in the Al Qasim region (within the Saq-Ram domain). An average natural recharge of 1.8 mm yr$^{-1}$ obtained by chloride mass-balance together with a mean measured water content on the vadose zone of 0.01% yields a local pore velocity equivalent to a 'natural recharge front velocity' of about 0.2 m yr$^{-1}$.

It is interesting to compare this recharge velocity with the water table decline velocity. By definition, this sand dune area is located away from any agricultural plot (i.e. zero artificial recharge by irrigation return flow) but within one of the largest drawdown areas worldwide (about 500 km diameter; Sharaf and Hussein, 1996) caused by intensive pumping. Considering a conservative 30 m water table decline in 45 years (BRGM and Abunayyan Trading Corp., 2008), a minimum 0.7 m yr$^{-1}$ decline is computed on the outskirts of this piezometric depression. This is significantly faster than the local natural recharge velocity of 0.2 m yr$^{-1}$, suggesting that the unsaturated zone is thickening faster than the percolation flows into it. Note that we estimated $(900 \pm 450) \times 10^6$ m$^3$ yr$^{-1}$ of irrigation return flow (2002-2019 average; see Sect. 3.2) corresponding to $(167 \pm 83)$ mm yr$^{-1}$ distributed only over the irrigated areas of the aquifer (i.e. about 5 400 km² for the 2002-2019 period; General Authority for statistics, 2019). Such a recharge value, two orders of magnitude larger than its natural counterpart, certainly prevents the disconnection between the recharge front and the free groundwater table in irrigated areas.

Hence, while irrigation excess is great enough to artificially sustain the recharge of the aquifer within agricultural plots, the effective recharge becomes locally and temporally zero on the outskirts of such crop areas, similar to observations of a semiarid aquifer of the North China Plain by Cao et al. (2016). Some regions behave as preferential recharge areas for the Saq-Ram Aquifer System, but a mechanism of a relative disconnection of the infiltration front with the declining water table likely occurs in intensively exploited regions, most probably in the vicinity of the main irrigated areas (represented by green areas in Figure 1) where there is no artificial recharge but still a piezometric drawdown induced by intensive pumping.

Therefore, the abstract, introduction and conclusion should be amended accordingly:

(Line 33) ~~. Within agricultural plots, irrigation excess is great enough to artificially recharge the aquifer (i.e. (167 ± 83) mm yr⁻¹ distributed over irrigated areas). However, on the outskirts of these crop areas subjected only to the natural recharge but still influenced by pumping drawdown, there is a risk of relative disconnection from the infiltration front with the declining water table (i.e. the unsaturated zone thickens faster than percolation flows through it), making effective recharge locally zero.~~

is replaced by: : chiefly induced by irrigation excess over irrigated surfaces (about 1% of the domain), artificial recharge corresponds to half of the total recharge of the aquifer.

(Lines 40-44) In order to improve the now shortened abstract, we suggest to add another discussion output regarding a methodological aspect of the GRACE approach, as recommended by the reviewer:

Due to large lag times of the diffuse recharge mechanism, annual analysis using this GRACE-GLDAS approach in arid domains should be limited to areas where focused recharge is the main mechanism, while long-term analysis is valid regardless of the recharge mechanism. Moreover, it appears that about 15 years of GRACE records are required to obtain a relevant long-term recharge estimate.

(Line 150)

is replaced by: this artificial recharge corresponds to half of the total aquifer recharge.

(Line 653)

is replaced by: Further, due to the intensive agricultural practices of the last decades, artificial recharge by irrigation excess (about 1% of the domain area), corresponds to half of the total recharge of the aquifer.

*1) At lines 156 & 157. Add "°C" after "27" and "8".*

Done (Lines 173-174).

*2) Substitute the last sentence of the caption of Figure 2, at lines 237 & 238, as follows: "Most of this previously published data comes from governmental entities, without providing any associated uncertainty."*

Done (Lines 253-254).